# Memory persistence and differentiation into antibody-secreting cells accompanied by positive selection in longitudinal BCR repertoires

**Artem Mikelov[1,2,3†], Evgeniia I Alekseeva[1†], Ekaterina A Komech[2,3], Dmitry B Staroverov[2], Maria A Turchaninova[2], Mikhail Shugay[2,3], Dmitriy M Chudakov[1,2,3], Georgii A Bazykin[1,4], Ivan V Zvyagin[2,3]***

[1]Skolkovo Institute of Science and Technology, Moscow, Russian Federation; [2]Shemyakin-Ovchinnikov Institute of Bioorganic Chemistry, Moscow, Russian Federation; [3]Institute of Translational Medicine, Pirogov Russian National Research Medical University, Moscow, Russian Federation; [4]A.A. Kharkevich Institute for Information Transmission Problems of the Russian Academy of Sciences, Moscow, Russian Federation

**Abstract** The stability and plasticity of B cell-mediated immune memory ensures the ability to respond to the repeated challenges. We have analyzed the longitudinal dynamics of immunoglobulin heavy chain repertoires from memory B cells, plasmablasts, and plasma cells from the peripheral blood of generally healthy volunteers. We reveal a high degree of clonal persistence in individual memory B cell subsets, with inter-individual convergence in memory and antibody-secreting cells (ASCs). ASC clonotypes demonstrate clonal relatedness to memory B cells, and are transient in peripheral blood. We identify two clusters of expanded clonal lineages with differing prevalence of memory B cells, isotypes, and persistence. Phylogenetic analysis revealed signs of reactivation of persisting memory B cell-enriched clonal lineages, accompanied by new rounds of affinity maturation during proliferation and differentiation into ASCs. Negative selection contributes to both persisting and reactivated lineages, preserving the functionality and specificity of B cell receptors (BCRs) to protect against current and future pathogens.

**\*For correspondence:**
izvyagin@gmail.com

†These authors contributed equally to this work

**Competing interest:** The authors declare that no competing interests exist.

## Editor's evaluation

By performing homeostatic longitudinal IgH repertoire analysis of human memory B cells and plasma cells, authors draw two major unique conclusions; first, a high degree of clonal persistence in individual memory B cell subsets with inter individual convergence in memory B cells and plasma cells; second, reactivation of persisting memory B cells with new rounds of affinity maturation during proliferation and differentiation into plasma cells. These conclusions provide significant insight into how human memory B and plasma cells are generated in a homeostatic condition.

## Introduction

B cells play a crucial role in protection from various pathogens and cancer cells as well as regulation of the immune response (*Akkaya et al., 2020*). The structural diversity of B cell receptors (BCRs) is responsible for the B cell-mediated immune system's capacity to recognize a wide variety of different antigens, and every individual harbors a large pool of naive B cell clones, each with

a unique BCR. Antigenic challenge triggers the proliferation and maturation of naive B cells with cognate BCRs, and the resulting progeny comprise a number of cell subsets with differing functions and lifespans. During the affinity maturation process, the initial structure of a given BCR can change at the genomic level as a result of somatic hypermutation (SHM), a process that accompanies B cell proliferation after antigen-specific activation. Cells bearing BCRs with higher affinity to the antigen are favored during the affinity maturation process, and produce signals that stimulate further differentiation and expansion (*De Silva and Klein, 2015*). Another process called class-switch recombination further increases the dimensionality of the BCR space. The five main classes, or isotypes, of antibodies (i.e., IgA, IgD, IgE, IgG, and IgM) have different functions in the immune response (*Stavnezer et al., 2008*; *Vidarsson et al., 2014*), and isotype switching during clonal proliferation can thereby change the functional capabilities of B cells and the antibodies they produce. As a consequence, antigen challenge yields a population of clonally related cells with different BCRs and functionalities.

Recently developed immune repertoire sequencing techniques provide valuable insights into the development and structure of B cell immunity with clonal-level resolution (*Chaudhary and Wesemann, 2018*). For example, the clonal relatedness of B cells in a given lineage, as well as the number and dynamics of B cell groups with distinct antigen specificities, can be studied based on BCR sequence homology. Numerous studies have generated valuable data by analyzing repertoire characteristics such as clonal diversity and tissue distribution, magnitude of clonal expansion and BCR SHM, V(D)J usage frequency and distribution of CDR3 length, and the degree of repertoire convergence and individuality (*Briney et al., 2019*; *Soto et al., 2019*; *Shah et al., 2018*; *Mandric et al., 2020*; *Yang et al., 2021*). Studies of BCR repertoires of patients with different diseases have made an important contribution to the understanding of mechanisms of pathology and B cell-mediated immunity (*Bashford-Rogers et al., 2019*; *Nielsen et al., 2020*; *Gaebler et al., 2021*; *Sakharkar et al., 2021*).

Longitudinal analysis of repertoires at different time points has made it possible to study the dynamics of B cell response following antigenic challenge or therapy (*Laserson et al., 2014*; *Davydov et al., 2018*; *Horns et al., 2019*; *Nourmohammad et al., 2019*; *Hoehn et al., 2021*). Reconstruction of BCR evolution in B cell clonal lineages and phylogenetic analysis can reveal which evolutionary forces predominate at different stages of clonal lineage development. De Bourcy et al. recently reported on age-related differences in the structure of clonal lineages, somatic hypermutagenesis and affinity maturation processes, and differences in recall response of persisting lineages upon vaccination depending on CMV seropositivity status (*de Bourcy et al., 2017*). Other studies have described in detail the action of positive selection in the evolution of clonal lineages in vaccination and chronic HIV infection (*Bonsignori et al., 2017*; *Horns et al., 2019*; *Nourmohammad et al., 2019*). Reports have also described persisting clonal lineages which are predominantly represented by cells with IgM/IgD isotypes, and which demonstrate signs of neutral evolution (*Horns et al., 2019*). Wu et al. observed the clonal stability of plasma cells (PL) in bone marrow (*Wu et al., 2010*), representing the largest fraction of ASCs in the human body. Comparison of BCR repertoires between different cell subsets also makes it possible to investigate factors governing the functional assignment of B cells during proliferation, and thereby to understand fundamental aspects of B cell immunity. For example, recent studies have described differences in BCR repertoires of IgM and switched memory B cells as well as the complex interplay between CD27$^{high}$ and CD27$^{low}$ B cell memory subsets, showing the complex nature of B cell immune memory (*Wu et al., 2010*; *Grimsholm et al., 2011*).

BCR repertoires of antigen-experienced B cell subsets and their dynamics are usually studied in the context of pathologic conditions and vaccination, and there is little equivalent data in the absence of acute or chronic immune response. We have therefore investigated immunoglobulin heavy chain repertoires from memory B cells, plasmablasts, and plasma cells from peripheral blood collected from generally healthy volunteers at three time points over the course of a year. In order to obtain detailed and unbiased repertoire data, we used advanced IgH repertoire profiling technology that provides full-length IgH variable region sequences with isotype annotation. Based on comparative and phylogenetic analysis of the resulting data, we are able to describe the structure, distinctive features, clonal relations, isotype distribution and temporal dynamics of B cell subset repertoires, as well as the phylogenetic history of large clonal lineages.

## Results

### IGH repertoire sequencing statistics and analysis depth

We collected peripheral blood from six healthy donors at three time points, where the second sample was collected 1 month after the first, and the third was collected 11 months after that (*Figure 1A*; *Table 1*). These samples were subjected to fluorescence-activated cell sorting (FACS) to isolate memory B cells (Bmem: CD19$^+$ CD20$^+$ CD27$^+$), plasmablasts (PBL: CD19$^{low/+}$ CD20$^-$ CD27$^{high}$ CD138$^-$), and plasma cells (PL: CD19$^{low/+}$ CD20$^-$ CD27$^{high}$ CD138$^+$; *Figure 1—figure supplement 1A*). Most of the cell samples were collected and processed in two independent replicates (*Figure 1—source data 1*). For each cell sample, we obtained IGH clonal repertoires using a 5'-RACE-based protocol, which allows unbiased amplification of full-length IGH variable domain cDNA while preserving isotype information, with subsequent unique molecular identifier (UMI)-based sequencing data normalization and error correction (*Turchaninova et al., 2016*; *Shugay et al., 2014*). From a total of 83 cell samples, we obtained $1.06 \times 10^7$ unique IGH cDNA molecules, each covered by at least three sequencing reads, representing $8.4 \times 10^5$ unique IGH clonotypes (*Figure 1—source data 1*). An IGH clonotype was defined as a unique nucleotide sequence spanning from the beginning of IGH V gene framework 1 to the 5' end of the C segment, sufficient to determine isotype. The number of unique clonotypes (i.e., species richness) depended on the number of cells per sample (*Figure 1—figure supplement 1B*), even after data normalization by sampling an equal number of unique IGH cDNA sequences. To characterize the number of distinct IGH clonotypes in each cell subset, we selected the samples with the most common number of sorted cells for each sample set. The median number of clonotypes was 20,072 (14,572–32,806, *n*=14) per $5 \times 10^4$ memory B cells, 628 (528–981, *n*=8) per $1 \times 10^3$ plasmablasts, and 800 (623–1183, *n*=9) per $1 \times 10^3$ plasma cells. Rarefaction analysis in the Bmem subpopulation (*Figure 1—figure supplement 1B*, left) revealed an asymptotic increase of species richness that did not reach a plateau, indicating that the averaged species richness can only serve as a lower limit of sample diversity estimation. For all samples of PBL and PL subpopulations, species richness curves plateaued, meaning that we had reached sufficient sequencing depth to evaluate the clonal diversity of the sorted cell samples (*Figure 1—figure supplement 1B*, center and right).

### B cell subsets display both divergent and similar characteristics in their IGH repertoires

First, we aimed to characterize features of the IGH repertoires of the Bmem, PBL, and PL subset based on several key properties: usage of germline-encoded IGHV segments, clonal distribution by isotypes, rate of SHM in CDR1-2 and FWR1-3, and features of the hypervariable CDR3 region. The proportion of overall clonal diversity occupied by the five major IGH isotypes was strikingly different between Bmem cells and antibody-secreting cells (ASCs; i.e., PBL and PL). IgM represented more than half of the repertoire in Bmem, while IgA was dominant in PBL and PL (*Figure 1B*, *Figure 1—source data 2*). The second most prevalent isotype in ASCs was IgG, which was also less abundant in Bmem compared to IgA. IgD represented a substantial part of the Bmem clonal repertoire, while <1% clonotypes of ASCs expressed IgD. The proportion of each isotype varied between donors and time points, but IgM and IgA or IgA and IgG consistently remained the most abundant isotypes in Bmem cells or ASCs, respectively (*Figure 1—figure supplement 2A*, *Figure 1—source data 2*). In all studied subsets, the isotype distribution in terms of number of unique clonotypes roughly mirrored the isotype distribution based on the number of IGH cDNA molecules, indicating absence of large clonal expansions or differences in IGH expression level distorting abundance of isotypes. This could not be determined by sequencing of bulk peripheral blood mononuclear cells (PBMCs), as higher levels of IGH expression by ASCs can change the isotype proportions and thereby bias the quantitation of clonotype abundance (*Figure 1—figure supplement 2B*). The obtained IGH isotype distributions based on unique clonotypes roughly correspond to the distribution of IGH isotypes typically detected by flow cytometry of the same subsets (*Perez-Andres et al., 2010*).

The level of SHM was on average significantly higher in ASC subsets, reflecting that PBLs and PLs are enriched for clones that have undergone affinity maturation (*Figure 1C*). The switched isotypes (IgG, IgA) had higher average levels of SHMs in the Bmem subset compared with IgM and IgD isotypes. Interestingly, the SHM level of IgD clonotypes in ASC subsets was significantly higher compared with Bmem. The average number of SHMs for IgE clonotypes did not differ significantly between cell subsets, but was significantly higher compared to the level of SHM detected for IgM

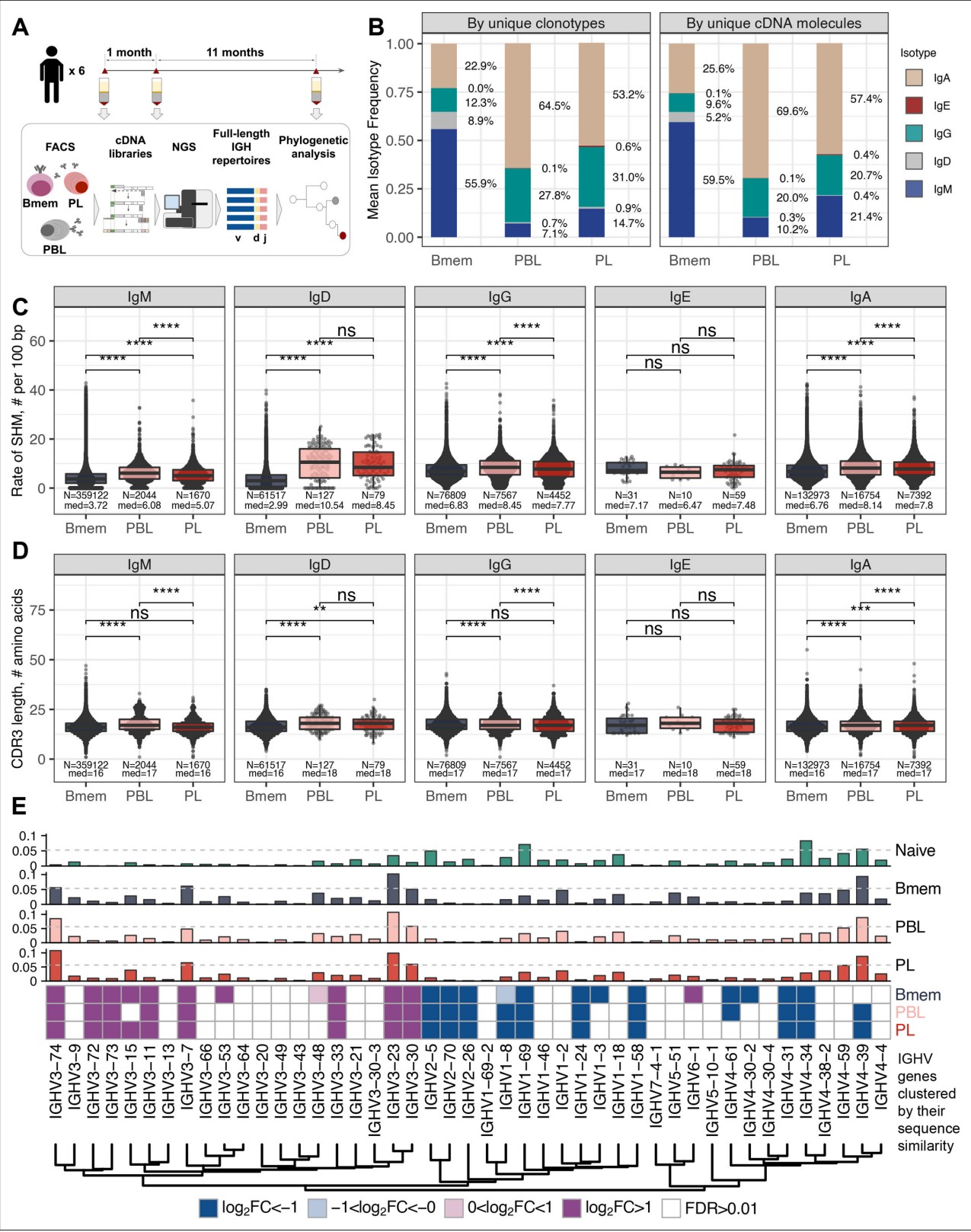

**Figure 1.** General characteristics of IGH repertoires in differentiated B cell lineage subsets. (**A**) Study design. Peripheral blood from six donors was sampled at three time points: T1 – initial time point, T2 – 1 month, and T3 – 12 months after the start of the study. At each time point, we isolated peripheral blood mononuclear cells (PBMCs) and sorted memory B cells (Bmem: CD19⁺ CD20⁺ CD27⁺), plasmablasts (PBL: CD19^low/+ CD20⁻ CD27^high CD138⁻), and plasma cells (PL: CD19^low/+ CD20⁻ CD27^high CD138⁺) in two replicates using fluorescence-activated cell sorting (FACS). For each cell sample,

*Figure 1 continued on next page*

*Figure 1 continued*

we obtained IGH clonal repertoires by sequencing respective cDNA libraries covering full-length IGH variable domain. (**B**) Proportion of isotypes in studied cell subsets averaged across all obtained repertoires. Left, frequency of unique IGH clonotypes with each particular isotype. Right, frequency of each isotype based on IGH cDNA molecules detected in a sample. (**C**) Distribution of the number of somatic hypermutations identified per 100 bp length of IGHV segment for clonotypes within each particular isotype. (**D**) Distribution of CDR3 length of clonotypes in each cell subset by isotype. (**E**) Distributions of average IGHV gene frequencies based on number of clonotypes in naive B cells (data from *Gidoni et al., 2019*), Bmem, PBL, and PL repertoires are shown at the top. Colored squares on heatmap indicate significantly different (false discovery rate, FDR < 0.01) frequencies for IGHV gene segments in corresponding B cell subsets compared to naive B cell repertoires. Color intensity reflects the magnitude of the difference (FC = fold change). Only V genes represented by more than two clonotypes on average are shown, data normalization was performed using trimmed mean of M values method (*Robinson and Oshlack, 2010*). IGHV gene segments are clustered based on the similarity of their amino acid sequence, as indicated by the dendrogram at the bottom. In C and D, the numbers at the bottom of the plots represent the number of clonotypes in the corresponding group, pooled from all donors, and the median measurements from each cell type. Comparisons between subsets were performed with two-sided Mann-Whitney U test. $*=p \leq 0.05$, $**=p \leq 0.01$, $***=p \leq 10^{-3}$, $****=p \leq 10^{-4}$.

The online version of this article includes the following source data and figure supplement(s) for figure 1:

**Source data 1.** Reperoire sequencing statistics for cell samples.

**Source data 2.** Isotype frequencies per cell sample.

**Figure supplement 1.** FACS gating strategy and rarefaction analysis of clonal diversity for memory B cells, plasmablasts and plasma cells.

**Figure supplement 2.** Isotype frequencies in studied subsets.

**Figure supplement 3.** The level of SHM in isotypes.

**Figure supplement 4.** IGHV gene frequencies in studied cell subsets.

and IgD clonotypes in Bmem (*Figure 1C*, *Figure 1—figure supplement 3*). Of note, the rate of SHM in PBLs was higher than that in PLs in clonotypes from the three most abundant isotypes (i.e., IgM, IgA, and IgG). We also compared the distributions of the lengths of the hypervariable CDR3 region between IGH clonotypes in different cell subsets. PBLs had significantly longer CDR3 regions compared to Bmem cells on average in every isotype except for IgE (*Figure 1D*). Of note, the average CDR3 length in PL clonotypes was significantly higher compared to Bmem for IgA and IgD, but not for the other isotypes.

IGHV gene segment usage was roughly similar between Bmem, PBL, and PL cells from all donors, indicating generally equal probabilities of memory-to-ASC conversion for B cells carrying BCRs encoded by distinct gene segments (*Figure 1E*, *Figure 1—figure supplement 4A*). This distribution differed significantly between the studied cell subsets and naive B cells (based on data from *Gidoni et al., 2019*). The repertoire of total B cells (Btot; CD19$^+$ CD20$^+$), which contained a large

**Table 1.** Donor demographics and cell sample sizes.

Multiple values in a cell separated by a semicolon represent replicates collected for the corresponding donor, time point, or cellular subset. AR – allergic rhinitis; FA – food allergy; HD – healthy donor.

| | | | | Number of cells per sample | | | | | | | | |
|---|---|---|---|---|---|---|---|---|---|---|---|---|
| Time point: | | | | T1 | | | T2 | | | T3 | | |
| Donor ID | Age | Sex | Status | Bmem | PBL | PL | Bmem | PBL | PL | Bmem | PBL | PL |
| D01 | 27 | F | AR | n/a | n/a | n/a | 50,300; 55,400 | 2100; 2100 | 1020; 1010 | 50,000; 50,000 | 1000; 1000 | 500; 500 |
| IM | 39 | M | AR,FA | 186,572 | 2200 | 129 | 69,900; 68,400 | 2000; 2486 | 920 | 50,000; 50,000 | 2000; 2000 | 1000; 1000 |
| MRK | 27 | M | AR | 143,162 | 5336 | 251 | 51,700; 50,600 | 2130; 2020 | 1000; 1035 | 50,000; 50,000 | 1000; 1000 | 400; 200 |
| AT | 23 | M | AR,FA | 101,400 | 7200 | 1,800 | 50,600; 57,400 | 2520 | 800 | 50000; 40800 | 1000; 1000 | 400; 200 |
| IZ | 33 | M | HD | 101,800 | 3900 | 850 | 50,500; 56,300 | 1140; 1840 | 1050; 625 | 50,000; 50,000 | 2000; 2000 | 200; 200 |
| MT | 33 | F | HD | n/a | n/a | n/a | n/a | n/a | n/a | 50,000; 50,000 | 1000; 1000 | 400 |

fraction of naive B cells, demonstrated similar IGHV gene segment usage to the naive B cell repertoire (*Figure 1—figure supplement 4A*). We observed statistically significant Pearson correlations in terms of IGHV gene frequencies for all pairs between Bmem, PBL, or PL (>0.95 correlation, p<0.01), and for naive vs. Btot (0.79 correlation, p<0.01). We observed high concordance in terms of under- or overrepresentation of specific IGHV gene segments in repertoires of all antigen-experienced B cell subsets compared to naive B cells; Pearson correlation coefficients for the fold-change of IGHV gene segment usage frequencies were 0.95 for Bmem and PBL, 0.96 for Bmem and PL, and 0.98 for PBL and PL (p<0.01 for all pairs). Moreover, IGHV gene segment under- or overrepresentation clearly depended on the given gene sequence. We clustered IGHV genes based on their sequence similarity, and observed that most IGHV segments in each of the four major clusters behaved concordantly with other segments in that cluster (*Figure 1E*). This effect was also observed at the level of individual repertoires (*Figure 1—figure supplement 4B*) with discrepancies that could probably be attributed to genetic polymorphism of the IGH loci of particular donors.

These observations highlight the differences in general characteristics of IGH repertoire between the Bmem and ASC subsets, and demonstrate similarity of IGHV gene usage that differs from that in naive B cells.

## Memory B cell repertoires are stable over time and contain a large number of public clonotypes

We further studied the similarity of IGH clonal repertoires of B cell subsets across time points and between individuals, evaluating repertoire stability (i.e., distance between different time points) and degree of individuality (i.e., distance between repertoires from different donors). We evaluated repertoire similarity at two levels of IGH sequence identity: frequency of clonotypes with identical nucleotide sequence-defined variable regions (FR1–4), and number of clonotypes with identical CDR3 amino acid sequences, IGHV gene segments, and isotypes. Both metrics showed significantly higher inter-individual differences compared to the divergence of repertoires derived from the same donor, reflecting the fact that IGH repertoires of Bmem, PBL, and PL subsets are private to a large degree (*Figure 2A and B*). We observed identical clonotypes in the repertoires of PBL and PL collected at different time points, whereas the repertoire similarity was much lower compared to that between replicate samples, reflecting the transient nature of PBL and PL populations in peripheral blood. Notably, we observed lower clonal overlap in PBL and PL for more distant time points (separated by 11 or 12 months) than those that are closer together (1 month) (*Figure 2—figure supplement 1A*). The dissimilarity between samples collected on the same day vs. 1 month or even 1 year later was much lower for Bmem, demonstrating the high stability of the clonal repertoire and long-term persistence of IGH clonotypes in these cells (*Figure 2B*, *Figure 2—figure supplement 1A*).

To better describe the inter-individual IGH repertoire convergence, we analyzed the number of IGH amino acid clonotypes shared between different donors (i.e., public clonotypes) among 5000 most expanded clonotypes in each Bmem repertoire, assuming that functional convergence could be detected among the most abundant clonotypes due to clonal expansions in response to common pathogens. Indeed, the average number of shared clonotypes in Bmem was significantly higher between fractions of the most abundant clonotypes compared to randomly sampled clonotypes (*Figure 2C*), as well as when compared to the most abundant clonotypes shared by two naive repertoires (from *Gidoni et al., 2019*) or to pre-immune IGH repertoires obtained by in silico generation using OLGA software (*Sethna et al., 2019*; *Figure 2C*). We also noted that there were no shared clonotypes defined by their full-length nucleotide sequence (*Figure 2—figure supplement 1B*). Public clonotypes were also hypermutated, although the rate of SHM was slightly lower compared to that in clonotypes specific to one donor (private) (*Figure 2—figure supplement 1C*). These observations indicate functional convergence in Bmem repertoires, which is presumably driven by exposure to common pathogens. Of note, the extent of clonal overlap was significantly higher between naive repertoires than for in silico-generated repertoires, indicating functional convergence even in pre-immune repertoires. Furthermore, the distance between V segment usage distributions in Bmem repertoires was not significantly different compared to that in naive B cells repertoires. That indicates that the higher clonotype sharing seen in Bmem cannot be attributed to lower diversity in IGHV germline usage (*Figure 2D*). The same analysis in PBL and PL subpopulations for the 600 and 200 most abundant clonotypes respectively yielded no shared clonotypes between repertoires of different

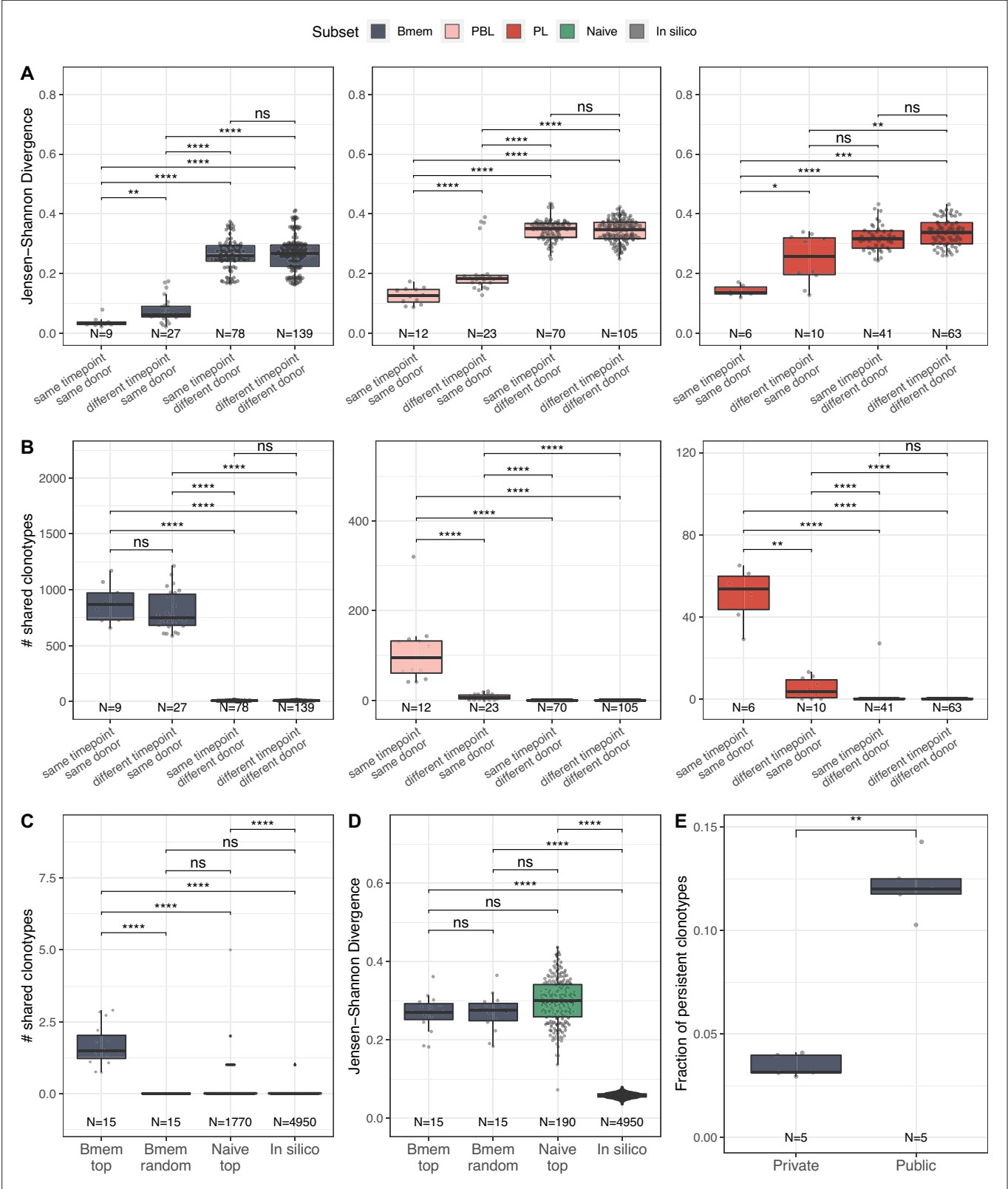

**Figure 2.** Memory B cells (Bmem), plasmablasts (PBL), and plasma cells (PL) IGH repertoire stability over time and similarity between individuals. (**A**) Distance between repertoires obtained at different time points from the same or different donors as calculated by Jensen-Shannon divergence index for IGHV gene frequency distribution. (**B**) Number of shared clonotypes between pairs of repertoires from the same or different donors and time points. For data normalization, we assessed the most abundant 14,000 Bmem, 600 PBL, and 300 PL clonotypes. (**C**) The average number of shared clonotypes

*Figure 2 continued on next page*

*Figure 2 continued*

between repertoires from pairs of unrelated donors for the most abundant Bmem clonotypes, randomly selected Bmem clonotypes, most abundant clonotypes from naive repertoires of unrelated donors (from *Gidoni et al., 2019*), or from synthetic repertoires generated with OLGA software; each repertoire in comparison was represented by a fixed number of clonotypes (5000), either most abundant, randomly selected, or generated where indicated. (**D**) Inter-individual distance between distributions of V genes in repertoires, calculated as Jensen-Shannon divergence indices for the pairs of repertoires depicted in C. (**E**) Fraction of persistent clonotypes detected at more than one time point among clonotypes detected in repertoires from only one donor (private) or in at least two donors (public). Each dot represents the fraction of persistent clonotypes from one donor. In all plots, clonotypes are defined as having identical CDR3 amino acid sequences and the same IGHV gene segment and isotype. For A–D, each dot represents a pair of repertoires of the corresponding type; N indicates the number of pairs of repertoires in the group. Comparisons in all panels were performed with two-sided Mann-Whitney U test. *=p ≤ 0.05, **=p ≤ 0.01, ***=p ≤ $10^{-3}$, ****=p ≤ $10^{-4}$.

The online version of this article includes the following figure supplement(s) for figure 2:

**Figure supplement 1.** IGH repertoire similarity within B cell lineage subpopulations.

donors, demonstrating no detectable convergence at this sampling depth. Finally, we found that public clonotypes were more likely to be detected than private ones in samples collected at different time points (*Figure 2E*), again suggesting persistent memory to common antigens. Thus, the results demonstrate the level of stability of memory BCR repertoires and extent of clonal sharing in repertoires of unrelated donors, which might be attributed to exposure to common antigens.

## Temporal dynamics of clonal lineages are associated with cell subset composition

SHM during BCR affinity maturation leads to the formation of clonal lineages – that is, BCR clonotypes evolved from a single ancestor after B cell activation. To study the structure and dynamics of clonal lineages originating from a single BCR ancestor, we grouped clonotypes from each individual based on their sequence similarity (see Materials and methods for details). We focused on larger clonal lineages consisting of at least 20 unique clonotypes from the corresponding donor. On average, these clonal lineages covered 3.4% of a given donor's repertoire, and we identified 190 such lineages across the four donors from whom samples were collected at each of the three time points (*Figure 3—figure supplement 1A*).

First we asked how B cell subsets and isotypes were represented in these most abundant clonal lineages. The clonal lineages were mostly composed of Bmem cell clonotypes of non-switched isotype IgM or were largely composed of ASCs, and enriched in IgG and IgA clonotypes (*Figure 3—figure supplement 1B*). To investigate the nature of such bimodal distribution and perform comparative analysis of these two types of clonal lineages, we divided them into two large clusters using *k*-means clustering algorithm, based on the proportion of represented cell subsets and BCR isotypes (*Figure 3A and B*, *Figure 3—figure supplement 2A*). The more abundant HBmem cluster included 138 clonal lineages, and was mostly composed of Bmem clonotypes of non-switched isotype IgM. Conversely, the smaller LBmem cluster (52 clonal lineages) was more diverse and largely composed of ASCs, and enriched in IgG and IgA clonotypes. The average size of clonal lineages (i.e., the number of unique clonotypes per lineage) did not differ between the HBmem and LBmem clusters (*Figure 3—figure supplement 2B*), and both clusters were present in repertoires of all donors (*Figure 3—figure supplement 2C*).

Next we tracked the abundance of each clonal lineage in the repertoire across each time point. The two clusters of lineages demonstrated different temporal behavior; while HBmem groups were quite stable over time, LBmem lineages had a burst of increased frequency at one of the time points (*Figure 3C*). To compare the temporal stability of clonal lineages, we defined the lineage persistence metric, which equals 1 when a clonal lineage was equally frequent at all three time points and is close to 0 when it was detected at just one time point (*Figure 3D*). Persistence of a clonal lineage was strongly associated with its composition (*Figure 3E and F*). Clonal lineages enriched with clonotypes or with the IgM isotype – including all HBmem lineages – were more likely to persist through time. Conversely, lineages with larger proportions of ASCs or IgG/IgA isotypes, including most LBmem lineages, tended to have lower persistence, with a burst of increased frequency at one particular time point. The time point of LBmem frequency burst varied between donors (*Figure 3—figure supplement 2D*). The frequencies of clonal lineages were highly correlated among replicate samples, and the

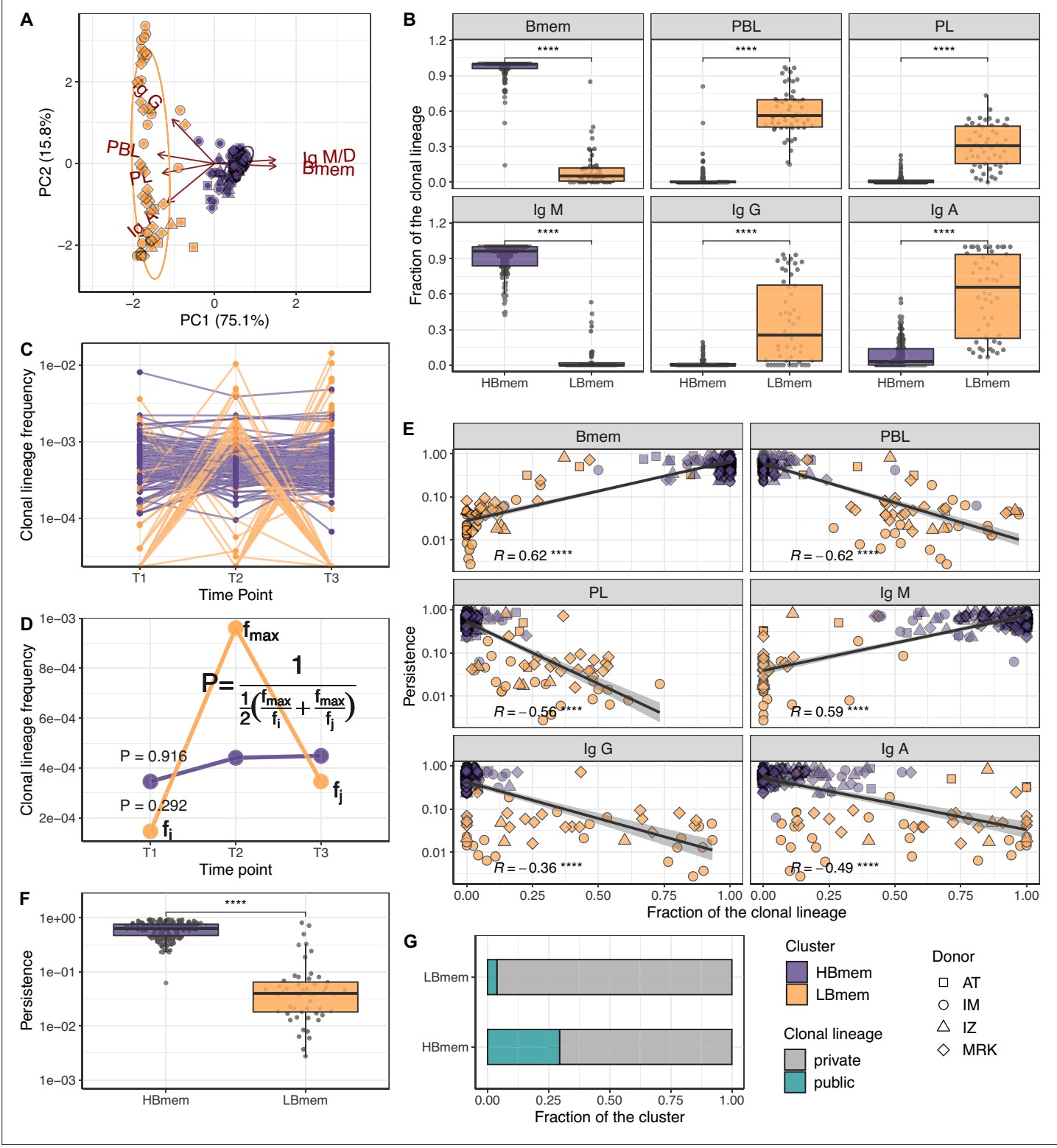

**Figure 3.** Temporal dynamics and composition of clonal lineages. (**A**) Principal component analysis (PCA) of clonal lineage composition: proportions of memory B (Bmem), plasmablasts (PBL), and plasma (PL) cells as well as proportions of isotypes. Arrows represent projections of the corresponding variables onto the two-dimensional PCA plane, with lengths reflecting how well the variable explains the variance of the data. The two principal components (PC1 and PC2) cumulatively explain 90.9% of the variance. Clonal lineages are colored according to the clusters they were assigned to by the *k*-means algorithm. (**B**) Proportion of clonotypes from the various cell subset or isotypes in clonal lineages falling into the HBmem or LBmem

*Figure 3 continued*

clusters. (**C**) Dynamics of clonal lineage frequency, defined as the number of clonotypes in a lineage divided by the total number of clonotypes detected at a given time point. Each line connects points representing a unique clonal lineage (*N*=190). (**D**) Schematic representation of how we calculated clonal lineage persistence. $f_{max}$ is the maximum clonal lineage frequency among the three time points, and $f_{i,j}$ are the frequencies at the remaining two time points. (**E**) Spearman's correlation between persistence of a clonal lineage and proportions of its clonotypes associated with a given B cell subset or isotype. (**F**) Comparison of persistence between HBmem and LBmem. (**G**) Fraction of public clonal lineages in the two clusters. Statistical significance for B, F, and G is calculated by the two-sided Mann-Whitney test. *=p ≤ 0.05, **=p ≤ 0.01, ***=p ≤ $10^{-3}$, ****=p ≤ $10^{-4}$.

The online version of this article includes the following figure supplement(s) for figure 3:

**Figure supplement 1.** Characteristics of the most abundant clonal lineages.

**Figure supplement 2.** Size, reproducibility and dynamics of HBmem and LBmem clonal lineages.

persistence of a clonal lineage was not associated with its size (*Figure 3—figure supplement 2E,F*), indicating that differences in persistence cannot be attributed to clonotype sampling noise.

Besides their higher persistence, the HBmem lineages were enriched in clonotypes detected at multiple time points (*Figure 3—figure supplement 2G*), indicating that persistent clonal lineages are supported by persistent clonotypes. Furthermore, 29.7% of the HBmem cluster was represented by public clonal lineages shared between at least two donors, compared to 3.8% for the LBmem cluster. The only two shared LBmem lineages had atypically high persistence, which made them more similar to HBmem (*Figure 3G*).

Thus, we observed two types of clonal lineages, representing different stages of an immune response: persisting memory with unswitched IgM isotype (HBmem) and responding lineages rapidly increasing in frequency and producing IgG or IgA antibodies (LBmem).

## LBmem clonal lineages could arise from HBmem clonal lineages

The evolutionary past of a clonal lineage can be described by inferring the history of accumulation of SHMs leading to individual clonotypes – that is, by reconstructing the phylogenetic tree of the clonal lineage. The initial germline sequence of each clonal lineage partially matches the germline VDJ segments, and can be reconstructed in a manner corresponding to the root of the phylogenetic tree of this lineage (see Materials and methods). However, the first node of the phylogenetic tree (green diamond in *Figure 4A*), the most recent common ancestor (MRCA) of the sampled part of the lineage, can be different from the inferred germline sequence. These differences, referred to as the G-MRCA distance, correspond to SHMs accumulated during the evolution of the clonal lineage prior to divergence of the observed clonotypes. The G-MRCA distance depends on how clonotypes of the tree were sampled. Sampling of clonotypes regardless of their position on the tree results in a low G-MRCA distance (*Figure 4A*, top panel), while sampling just those clonotypes belonging to a particular clade can conceal early stages of lineage evolution and thus result in a large G-MRCA distance (*Figure 4A*, bottom panel).

The G-MRCA distance was on average fivefold higher in LBmem clonal lineages compared to HBmem (median = 0.044 vs. 0.008, *Figure 4B*). This means that even though nearly all the evolution of an HBmem clonal lineage leaves a trace in the observed diversity of that lineage (*Figure 4D and G*), the sequence variants of an LBmem lineage typically result from divergence of an already-hypermutated clonotype (*Figure 4E and H*). In most (38 out of 52) LBmem lineages, some Bmem clonotypes were observed at the time point preceding expansion. Moreover, clonotypes of LBmem lineages are typically characterized by lower pairwise divergence compared to that in HBmem lineages (median = 0.11 vs. 0.13, *Figure 4C*). Together with the burst-like dynamics characteristic of LBmem lineages (*Figure 3F*), this implies that LBmem lineages may represent recent, rapid clonal expansion of pre-existing memory.

Based on these results and the compositional features of the two clusters, we further hypothesized that LBmem clonal lineages may arise from reactivation of pre-existing memory cells belonging to the HBmem cluster. In search of examples of such a transition, we examined all clonal lineages that were persistent but included ASC clonotypes. We found one clear example of a transition from HBmem to LBmem state in the evolutionary history of a clonal lineage (*Figure 4F,I*). While the MRCA of this lineage nearly matched the germline sequence, all ASC clonotypes were grouped in a single monophyletic clade (sublineage), such that its ancestral node was remote from the MRCA. The ASC sublineage demonstrated all features characteristic of LBmem, including predominance of IgG and

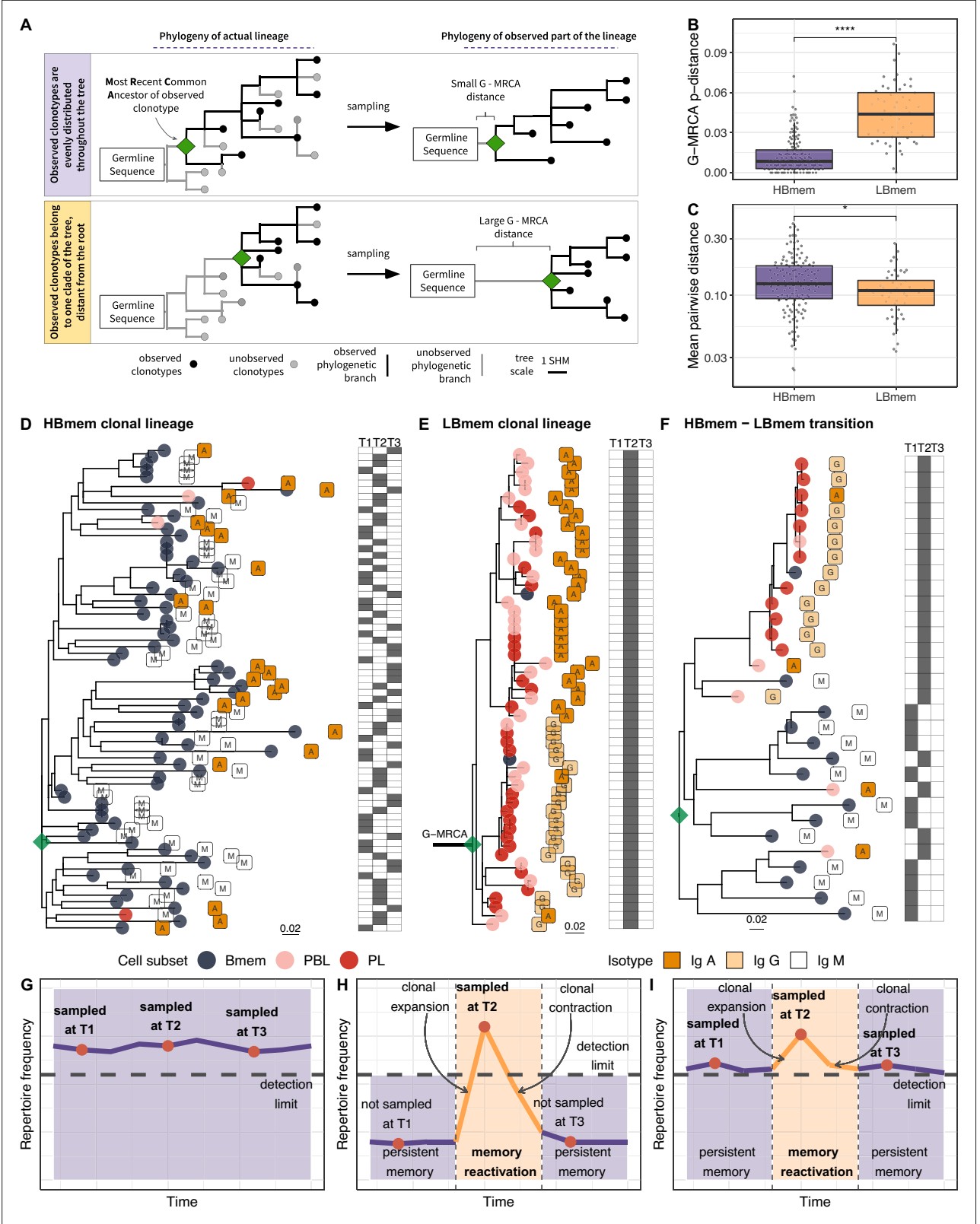

**Figure 4.** Phylogenetic history of HBmem and LBmem clonal lineages. (**A**) A schematic illustration of how the distances between the germline sequence and the most recent common ancestor (MRCA) of a clonal lineage (G-MRCA distance) vary depending on which subset of clonotypes is sampled: a sample uniform with regard to the position on the tree (top panel), or only those belonging to a particular clade of the tree (bottom panel). (**B**) Comparison of G-MRCA p-distance (i.e., the fraction of differing nucleotides) for HBmem and LBmem lineages. (**C**) Mean pairwise phylogenetic

*Figure 4 continued on next page*

*Figure 4 continued*

distance (i.e., the distance along the tree) between clonotypes of the same lineage for HBmem and LBmem clusters. (**D–F**) Representative phylogenetic trees for clonal lineages belonging to HBmem (**D**), LBmem (**E**), and an example of HBmem-LBmem transition (**F**). The LBmem sublineage in F is nested deep in the phylogeny of the memory clonotypes, and is not characterized by a particularly long ancestral branch, indicating that it is not an artifact of clonal lineage assignment. Circles correspond to individual clonotypes, with the cellular subset indicated by color, and the isotype by label. Tables at right indicate the presence or absence of the corresponding clonotype at each time point. The G-MRCA distance is indicated with a thick line. (**G–I**) Schematic representation of the hypothetical dynamics of relative size for clonal lineages represented in D, E, and F, respectively. Significance for B and C was obtained by the two-sided Mann-Whitney test. *=p ≤ 0.05, **=p ≤ 0.01, ***=p ≤ $10^{-3}$, ****=p ≤ $10^{-4}$.

The online version of this article includes the following figure supplement(s) for figure 4:

**Figure supplement 1.** Analysis of the sequence of clonotypes comprising the lineage with HBmem-LBmem the transition.

IgA isotypes, low persistence, and low clonotype divergence. Conversely, the remainder of the clonal lineage had features of HBmem: predominance of IgM, high persistence, and high levels of clonotype divergence. Position of ASC sublineage on a distant node from the root of the tree indicates gradual accumulation of SHMs, distinguishing the ASC sublineage from the remaining clonotypes. This fact together with the similarity of CDR3 regions of lineage clonotypes (*Figure 4—figure supplement 1*) give a reason to conclude that the ASC sublineage has the same origin as the remaining part of the tree with features of HBmem cluster.

To summarize, we observed that LBmem lineages had low level of clonotype divergence and large distance of lineage's ancestor from the germline sequence, assuming their recent origin from a mature clonotype. The temporal dynamics of LBmem, detection of Bmem clonotypes at the time point prior to the LBmem lineage expansion, and the relationship between HBmem and LBmem on a clonal lineage level suggest that LBmem expansions may result from reactivation of pre-existing memory.

## Reactivation of LBmem clonal lineages is driven by positive selection

Having shown that the LBmem lineages likely originate from clonal expansion of pre-existing memory, we further compared the contribution of positive (favoring new beneficial SHMs) and negative (preserving the current variant) selection between the LBmem and HBmem clusters. Since we observed only one clear example of an HBmem-LBmem transition (*Figure 4F*, *Figure 4—figure supplement 1*), we could not claim with certainty that LBmem lineages always emerge from pre-existing HBmem lineages rather than from some other memory type. Still, we were able to study LBmem reactivation by comparing differences in substitution patterns at the origin of HBmem and LBmem clusters. We reasoned that the G-MRCA distance of an HBmem lineage contains mutations fixed by primary affinity maturation after the initial lineage activation. In contrast, the G-MRCA distance of an LBmem lineage contains both mutations arising during primary affinity maturation and subsequent changes occurring later in the evolution of the lineage. Differences in the characteristics of the G-MRCA mutations between clusters are therefore informative of the process prior to the observed expansion of LBmem lineages.

To assess selection at the origin of the HBmem and LBmem lineages, we measured the divergence of nonsynonymous sites relative to synonymous sites (i.e., the DnDs ratio). Classically, DnDs > 1 is interpreted as evidence for positive selection. However, DnDs > 1 is rare, because the signal of positive selection is usually swamped by that of negative selection. In the McDonald-Kreitman (MK) framework, positive selection is instead revealed by excessive nonsynonymous divergence relative to nonsynonymous polymorphism (i.e., DnDs > PnPs; see Materials and methods and *Figure 5—source data 1* for examples), under the logic that advantageous changes contribute more to divergence than to polymorphism (*McDonald and Kreitman, 1991*). The fraction of adaptive nonsynonymous substitutions ($\alpha$) can then be estimated from this excess. We designed an MK-like analysis, comparing the relative frequencies of nonsynonymous and synonymous SHMs at the G-MRCA branch (equivalent to divergence in the MK test) to those in subsequent evolution of clonal lineages (equivalent to polymorphism in the MK test; *Figure 5A*, see Materials and methods).

In both the HBmem and LBmem clonal lineages, we observed a higher ratio of nonsynonymous to synonymous SHMs in the G-MRCA branches compared to subsequent tree branches, meaning that a fraction of SHMs acquired by MRCA was further fixed by positive selection. However, this fraction was higher in LBmem lineages (Fisher's exact test: $\alpha$=0.58 and 0.65, p<$10^{-6}$ and <$10^{-15}$ in HBmem and LBmem, respectively). $\alpha$ of distinct clonal lineages was also generally higher in LBmem than in

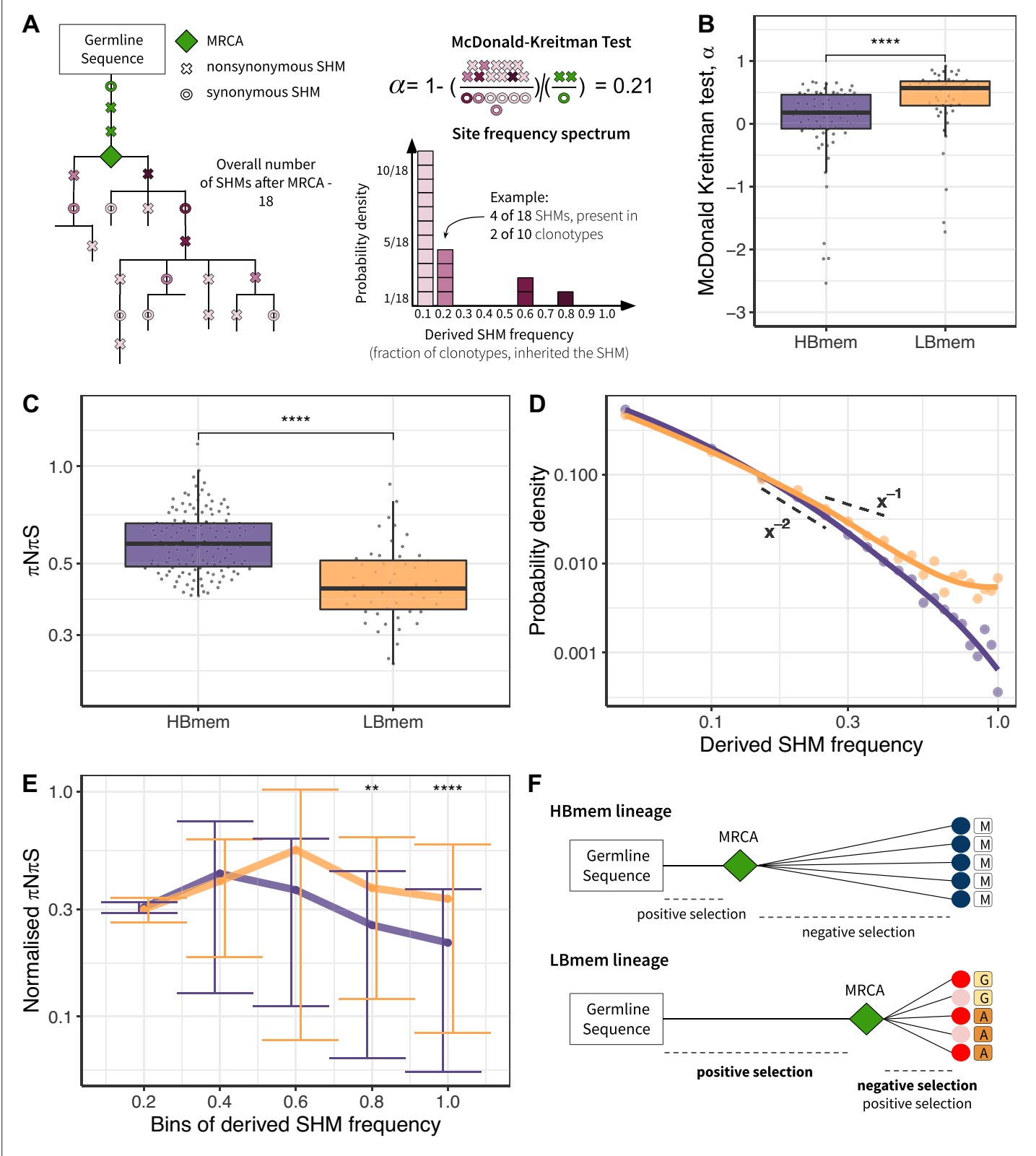

**Figure 5.** Signatures of positive and negative selection in HBmem and LBmem clusters. (**A**) Schematic of the McDonald-Kreitman (MK) test and site frequency spectrum (SFS) concept. (**B**) MK estimate of the fraction of adaptive nonsynonymous changes (α) between germline and most recent common ancestor (MRCA) in HBmem and LBmem clonal lineages. Only lineages with nonzero G-MRCA distance are included. *N*=68 for HBmem, 49 for LBmem, see **Figure 5—source data 2**. (**C**) Comparison of mean pairwise *πNπS* of HBmem and LBmem lineages. (**D**) Averaged SFS for HBmem and LBmem

*Figure 5 continued on next page*

*Figure 5 continued*

clonal lineages. The two dashed lines correspond to $f(x)=x^{-1}$, which is the expected neutral SFS under Kingman's coalescent model (Kingman 1982), and $f(x)=x^{-2}$. (**E**) Comparison of normalized $\pi N \pi S$ for HBmem and LBmem clonal lineages in various SHM frequency bins. The number of polymorphisms in each bin is normalized to the overall number of polymorphisms in a corresponding clonal lineage. (**F**) Scheme summarizing features of HBmem and LBmem clonal lineages. Comparisons in B, C, and E were performed by two-sided Mann-Whitney test, with Bonferroni-Holm multiple testing correction in E. *=p ≤ 0.05, **=p ≤ 0.01, ***=p ≤ $10^{-3}$, ****=p ≤ $10^{-4}$.

The online version of this article includes the following source data for figure 5:

**Source data 1.** Examples of divergent and polymorphic sites as calculated for the McDonald-Kreitman test.

**Source data 2.** MK test results under different inclusion criterion for clonal lineages from HBmem and LBmem clusters.

HBmem (median $\alpha$=0.57 vs. $\alpha$=0.18, *Figure 5B*), showing that positive selection more frequently preceded expansion of LBmem than HBmem lineages. The observation of excess $\alpha$ in the LBmem cluster compared to HBmem was robust to the peculiarities of the MK analysis (*Figure 5—source data 2*). The higher $\alpha$ for LBmem compared to HBmem implies that a larger fraction of SHMs was positively selected in LBmem clonal lineages before their expansion. This excess of advantageous SHMs in ancestors of LBmem lineages together with previous observations that LBmem lineages can originate from reactivated memory suggests that reactivation was coupled with new rounds of affinity maturation.

## Subsequent evolution of LBmem clonal lineages is affected by negative and positive selection

Next, we considered the effects of selection on HBmem and LBmem clusters following their divergence from their MRCAs that is, in the subsequent evolution of a clonal lineage leading to the diversity of the observed clonotypes. We calculated the per-site ratio of nonsynonymous and synonymous SHMs ($\pi N \pi S$) among those that originated after the MRCA. The $\pi N \pi S$ of both clusters was <1 (*Figure 5C*). This deficit of nonsynonymous SHMs indicates negative selection in the observed part of clonal lineage evolution. The $\pi N \pi S$ ratio was lower in the LBmem cluster, indicating stronger negative selection.

To examine the selection affecting these post-MRCA SHMs in more detail, we studied the frequency distribution of SHMs in individual lineages, or their site frequency spectra (SFSs) (*Nielsen, 2005*; *Neher and Hallatschek, 2013*; *Nei and Kumar, 2000*; *Horns et al., 2019*; *Nourmohammad et al., 2019*; *Figure 5A*). SFS reflects the effect of selection on these SHMs. Deleterious SHMs are held back by negative selection, so that their frequency in the lineage remains low. By contrast, positive selection favors the spread of adaptive SHMs, increasing their frequency. Therefore, negative selection biases the SFS toward low frequencies, and positive selection, toward high frequencies. For each clonal lineage, we reconstructed the SFS of the SHMs accumulated since divergence from MRCA (*Figure 5A*), and then averaged these SFSs within the HBmem and LBmem clusters. A larger proportion of the LBmem SFS corresponds to high frequencies compared to HBmem (*Figure 5D*), indicating weaker negative and/or stronger positive selection in LBmem SFS.

To distinguish between these selection types, we calculated the proportion of the SFS distribution falling into each frequency bin for nonsynonymous SHMs, and divided it by the same value for synonymous SHMs (normalized $\pi N \pi S$; see Materials and methods, *Figure 5E*). The inter-cluster differences in the normalized $\pi N \pi S$ in low-frequency bins were generally reflective of negative selection, while the differences in the high-frequency bins were reflective of positive selection. Normalized $\pi N \pi S$ was significantly higher in the high-frequency (>60%) bins of SHMs in LBmem clonal lineages. This indicates that for LBmem, those nonsynonymous changes that were not removed by negative selection reached high frequencies more often than in HBmem. In total, these data indicate that a fraction of nonsynonymous mutations accumulated by LBmem lineages were adaptive. We thus observed that reactivation of LBmem lineages is coupled with strengthening of both types of selection: positive on the G-MRCA branch, and both positive and negative during subsequent clonal lineage expansion. This pattern is most likely evidence of new rounds of affinity maturation, which result in the acquisition of new advantageous changes and preserve the resulting BCRs from deleterious ones. HBmem, in contrast, evolved more neutrally under weaker negative selection, suggesting absence of antigen challenge during the observation period (*Figure 5F*).

# Discussion

Using advanced library preparation technology, we performed a longitudinal study of BCR repertoires of the three main antigen-experienced B cell subsets – memory B cells, plasmablasts, and plasma cells – from peripheral blood of six donors, sampled three times over the course of a year. We analyzed these repertoires from two conceptually different but complementary points of view. First, we compared various repertoire features between the cell subsets, including clonotype stability in time and convergence between individuals. Second, we tracked the most abundant B cell clonal lineages in time and analyzed their cell subset and isotype composition, phylogenetic history, and mode of selection.

Comparative analysis of the cell subsets revealed significant differences in IGH isotype distribution, rate of SHM, and CDR3 length. IgM clonotypes predominated in the Bmem subset, whereas in ASCs the switched isotypes IgA and IgG together represented >80% of repertoire diversity on average. As expected, classical switched isotypes have higher rates of SHM, and the rate of SHM in ASCs is in general higher than in Bmem. The IgD isotype in Bmem cells showed similarities to IgM, where most IgD clonotypes had low levels of SHM, although there was a fraction of heavily mutated clonotypes. On average, IgD-switched PL and PBL had a comparable level of SHM with IgG- and IgA-expressing ASC clonotypes. Notably, the level of SHM and CDR3 length in PBL on average exceeded that of PL in IgM, IgA, and IgG isotypes. We hypothesize that such PBLs with heavily hypermutated BCRs could be the subset of B cell progeny that continue to acquire mutations after optimal affinity has been achieved, while another part of the clonal progeny is committed to a long-lived PL fate and acquires the CD138 marker characteristic of this cell subset (*Garimalla et al., 2019*).

While different in many aspects, immune-experienced B cell subsets are similar – and concordantly distinct from naive B cells – in terms of IGHV gene segment usage. Moreover, we observed that the correlated enrichment or depletion in V segment usage frequency generally coincides with the level of sequence similarity of the V segments. Most IGHV-3 family members were observed more frequently in antigen-experienced B cells compared to naive subsets in all donors and time points, while most of the other V genes that are well represented in the naive subset decreased in frequency. These differences in V usage frequencies between naive and antigen-experienced B cell subsets have also been reported in several previous studies, even though different FACS gating strategies were used (*Mitsunaga and Snyder, 2020*; *Ghraichy et al., 2021*). Our findings further support the idea that initial recruitment of B cells to the immune response is in many cases determined by the germline-encoded parts of the BCR, presumably CDR1 and CDR2. Previous studies have shown high levels of convergence in IGHV usage between B cell clonotypes specific for particular pathogens or self-antigens (*Peng et al., 2019*; *Galson et al., 2015*; *Bashford-Rogers et al., 2019*).

We further analyzed the repertoire similarity of cell subsets over time and between individuals. Intuitively, the Bmem subset is the most stable over time, showing less repertoire divergence and a greater number of shared clonotypes between sampling time points in the same individuals. Our finding expands the recent observation of Bmem subset stability in elderly donors (*Phad et al., 2022*) on a larger cohort of donors of younger age. Compared to intra-individual sharing, we detected a very small number of common clonotypes in Bmem cells. Those clonotypes have comparable levels of SHM to private ones, assuming a germinal center-dependent origin. Two recent studies on extra-deep repertoires of bulk peripheral blood B cells reported 1–6% (*Soto et al., 2019*) or ~1% (*Briney et al., 2019*) shared V-CDR3aa-J clonotypes between pairs of unrelated donors, with lower repertoire convergence for class-switched clonotypes shown in the latter study. Using the same method, we similarly measured 0.06% repertoire overlap in the Bmem subset (*Figure 2—figure supplement 1D*). Complementing the model proposed by Briney et al. – wherein IGH repertoires are initially dissimilar and then homogenize during B cell development before finally becoming highly individualized after immunological exposure – we found a significantly higher number of shared clonotypes between IGH repertoires among the most abundant Bmem clonotypes, indicating functional convergence presumably due to exposure to common environmental antigens. The latter is further supported by the higher number of persisting Bmem clonotypes observed among public clonotypes compared to private ones.

Next, we focused on the most abundant B cell clonal lineages, which are large enough to study the interconnection between cell subsets and phylogenetic features of lineages. In all individuals, the observed clonal lineages clearly fell into two clusters. HBmem represents persistent memory with a

predominant IgM isotype; such clonal lineages were equally sampled from all time points and rarely included ASC clonotypes. The MRCA of observed clonotypes in HBmem lineages almost matched the predicted germline sequence – and in 14.5% of the lineages, matched completely – indicating that the probability of observing a clonotype from these lineages has no association with the position in that lineage's phylogeny. Horns and colleagues observed lineages with very similar features to HBmem, which also possessed persistent dynamics against a background of vaccine-responsive lineages and were predominantly composed of the IgM isotype (*Horns et al., 2019*). However, their study was performed on bulk B cells, so there was no possibility to track their relatedness to the Bmem subset. In contrast, the LBmem cluster demonstrates completely different features, with lineages largely composed of ASC clonotypes with switched IgA or IgG isotypes, showing active involvement in ongoing immune response. The MRCA of LBmem lineages differed from the germline sequence by some number of SHMs, and only 1.9% of LBmem lineages had a complete match between the MRCA and the germline sequence. A large G-MRCA distance implies that the observed clonotypes originated from an already-hypermutated ancestor, and that we had therefore sampled clonotypes from a single clade of the lineage phylogeny. Such an effect can be caused by both rapid expansion of the clade and migration of the clade's clonotypes, diverged in the tissue of residence (*Mandric et al., 2020*). We also observed that most LBmem lineages expanded at T2 or T3 (38 out of 45, >80%) had at least one clonotype detected in the Bmem subset at the previous time point, leading us to conclude that LBmems represent the progeny of reactivated Bmem cells. We found one clear example that further supports this idea: a lineage that possesses all features of the HBmem cluster except for one monophyletic clade, typical for LBmem lineage. This example of HBmem-LBmem transition is very similar to reactivated persistent memory, as observed by Hoehn et al. in response to seasonal flu vaccination (*Hoehn et al., 2021*). In addition, Phad et al. have recently demonstrated clonal relatedness of the emerging PBL to the persistent Bmem lineages in longitudinal immune repertoire profiling of aged healthy donors (*Phad et al., 2022*). Thus, it can be assumed that at least part of the observed LBmem lineages are the progeny of the persistent memory represented by HBmem lineages.

Our analysis of the selection mode in HBmem and LBmem lineages supported our assumptions. We showed that both lineages experienced positive selection from the germline sequence to the MRCA of the observed clonotypes – as expected, assuming that primary B cell activation is followed by affinity maturation associated with clonal lineage expansion. However, the pressure of positive selection was stronger in LBmem lineages than in HBmem. In addition, we detected an excess of sites under positive selection in LBmem lineages that underwent evolution after the MRCA as well. This leads us to the hypothesis that LBmem cells underwent additional rounds of affinity maturation after reactivation. Hoehn et al. did not study the mode of selection in their reactivated lineages, but some clonotypes were sampled from germinal centers, supporting the involvement of affinity maturation in the process of memory reactivation. In subsequent evolution after the MRCA, we detected negative selection in both types of lineages – but again, stronger in LBmem. This excessive negative selection in LBmem lineages can be considered as a signature of purification of the clonal lineage from deleterious BCR variants during affinity maturation.

In the present study we focused on the three major antigen-experienced cell subsets defined by the set of cell surface markers, while further validation using more advanced techniques for cell phenotyping (e.g., scRNA-seq) is desirable. Peripheral blood as the source of cell samples excludes from the analysis the cells resident in different tissues, such as bone marrow niches, the main source of long-living PL cells in humans. Also, we focused our analysis on the most expanded clonotypes and clonal lineages. These aspects limit our findings to some extent, meaning that it can reflect only a part of the whole complex picture of B cell immunity behavior in a normal state. The number of donors we studied was relatively small and the cohort was combined from donors different in allergy status. Nevertheless, the whole dataset was large enough to reveal that all our observations are significant and stay reproducible among donors independent from their health conditions. We observed no evidence that allergy status affects the structure of our data (Appendix 1), which allowed us to generalize obtained observations for the whole cohort group. We have not observed much direct evidence of the process of memory reactivation and new rounds of affinity maturations. Reactivation process was clearly detected in only one clonal lineage (*Figure 4F*). However, this explanation of the given data is convincing because of the whole set of indirect evidence, such as large G-MRCA distance and close relatedness of LBmem clonotypes, the presence of Bmem clonotypes prior to LBmem expansion, and

different modes of natural selection in HBmem and LBmem clusters. Our hypothesis is also supported by recent studies (*Hoehn et al., 2021*; *Phad et al., 2022*).

Thus, in this work, we performed a detailed longitudinal analysis of BCR repertoires from immune-experienced B cell subsets from donors without severe pathologies, and from these data, we have produced a framework for the comprehensive analysis of selection in BCR clonal lineages. Our results demonstrate the interconnection of B cell subsets at a clonal level, B cell memory convergence in unrelated donors, and the long-term persistence of memory-enriched clonal lineages in peripheral blood. Signs of positive selection were detected in both memory- and ASC-dominated B cell lineages. Together, the results of our evolutionary analysis of B cell clonal lineages coupled with B cell subset annotation suggest that the reactivation of pre-existing memory B cells is accompanied by new rounds of affinity maturation.

# Materials and methods

**Key resources table**

| Reagent type (species) or resource | Designation | Source or reference | Identifiers | Additional information |
|---|---|---|---|---|
| Antibody | Anti-CD19-APC (mouse monoclonal) | Miltenyi Biotec | clone: LT19, cat. #:130-091-248 | FACS (2 µl per test) |
| Antibody | Anti-CD20-VioBlue (mouse monoclonal) | Miltenyi Biotec | clone: LT20, cat. #:130-094-167 | FACS (2 µl per test) |
| Antibody | Anti-CD27-VioBright FITC (mouse monoclonal) | Miltenyi Biotec | clone: M-T271, cat. #:130-104-845 | FACS (2 µl per test) |
| Antibody | Anti-CD138-PE-Vio770 (mouse monoclonal) | Miltenyi Biotec | clone: 44F9, cat. #:130-099-292 | FACS (2 µl per test) |
| Software, algorithm | MIGEC | *Shugay et al., 2014* | v1.2.7 | |
| Software, algorithm | MiXCR | *Bolotin et al., 2015* | v3.0.10 | |
| Software, algorithm | TIgGER | *Gadala-Maria et al., 2015* | v3.0.10 | |
| Software, algorithm | Change-O | *Gupta et al., 2015* | v0.4.4 | |
| Software, algorithm | edgeR | *Robinson et al., 2010* | v0.4.4 | |
| Software, algorithm | MUSCLE | *Edgar, 2004* | v3.8.31 | |
| Software, algorithm | RAxML | *Stamatakis, 2014* | v8.2.11 | |
| Software, algorithm | R language | *R Development Core Team, 2018* | v4.0.0 | |
| Software, algorithm | ggplot2 | *Ginestet, 2011* | v3.3.2 | |
| Software, algorithm | ggtree | *Yu et al., 2017* | v2.2.4 | |

## Donors, cells, and time points

Blood samples from six (four males and two females) young and middle-aged donors (23, 27, 27, 33, 33, and 39 years of age) without severe inflammatory diseases, chronic or recent acute infectious diseases, or vaccinations were collected at three time points (T1 – 0, T2 – 1 month, T3 – 12 months); donor details and the number cells collected for each time point and cell subset are provided in *Table 1*. Four donors suffered allergic rhinitis to pollen, and two also suffered from food allergy. Informed consent was obtained from each donor. The study was approved by the Local Ethical Committee of Pirogov Russian National Research Medical University, Moscow, Russia (abstract #190 November 18, 2019). At each time point, 18–22 ml of peripheral blood was collected in BD Vacuette tubes with EDTA. PBMCs were isolated using Ficoll gradient density centrifugation. To isolate subpopulations of interest, cells were stained with anti-CD19-APC, anti-CD20-VioBlue, anti-CD27-VioBright FITC, and anti-CD138-PE-Vio770 (all Miltenyi Biotec) in the presence of FcR Blocking Reagent (Miltenyi Biotec) according to the manufacturer's protocol, and then sorted using FACS (BD FacsAria III, BD Biosciences) into the following populations: Bmem cells (CD19$^+$ CD20$^+$ CD27$^+$ CD138$^-$), PBL (CD20$^-$ CD19$^{Low/+}$ CD27$^{++}$ CD138$^-$), PL cells (CD20$^-$ CD19 $^{Low/+}$ CD27$^{++}$ CD138$^+$). For each donor at T1, one replicate sample of

each cell subpopulation was collected. At T2 and T3, two replicate samples were collected ($50 \times 10^3$ to $100 \times 10^3$ Bmem, $1 \times 10^3$ to $2 \times 10^3$ PBL, $0.5 \times 10^3$ to $1 \times 10^3$ PL per sample).

## IGH cDNA libraries and sequencing

IGH cDNA libraries were prepared as described previously (*Turchaninova et al., 2016*) with several modifications. Briefly, we used a rapid amplification of cDNA ends (RACE) approach with a template-switch effect to introduce 5′ adaptors during cDNA synthesis. These adaptors contained both UMIs, allowing error correction, and sample barcodes (described in *Zvyagin et al., 2017*), allowing us to rule out potential cross-sample contaminations. In addition to a universal sequence for annealing the forward PCR primer, we also introduced a 5′ adaptor during the reverse transcription (RT) reaction, which allowed us to avoid using multiplexed forward primers specific for V segments, thereby reducing PCR amplification biases. Multiplexed C-segment-specific primers were used for RT and PCR, allowing us to preserve isotype information. Prepared libraries were then sequenced with an Illumina HiSeq 2000/2500 (paired-end, 2×310 bp).

## Sequencing data pre-processing and repertoire reconstruction

Sample demultiplexing by sample barcodes introduced in the 5′ adapter and UMI-based error correction were performed using MIGEC v1.2.7 software (*Shugay et al., 2014*). For further analysis, we used sequences covered by at least two sequencing reads. Alignment of sequences, V-, D-, J-, and C-segment annotation, and reconstruction of clonal repertoires were accomplished using MiXCR v3.0.10 (*Bolotin et al., 2015*) with prior removal of the primer-originated component of the C-segment. We defined clonotypes as a unique IGH nucleotide sequence starting from the framework 1 region of the V segment to the end of the J segment, and taking into account isotype. Using TIgGER (*Gadala-Maria et al., 2015*) software, we derived an individual database of V gene alleles for each donor and realigned all sequences for precise detection of hypermutations. For analysis of general repertoire characteristics (isotype frequencies, SHM levels, CDR3 length, IGHV gene usage, and repertoire similarity metrics), we used samples covered by at least 0.1 cDNA molecules per cell for Bmem, and at least five cDNA per cell for PBL and PL.

## Repertoire characteristics analysis

Isotype frequencies, rate of SHM, and CDR3 lengths were determined using MiXCR v3.0.10 (*Bolotin et al., 2015*). For calculation of background IGHV gene segment usage and number of shared clonotypes, we utilized data derived from *Gidoni et al., 2019* (European Nucleotide Archive accession number ERP108501) representing naive B cell IGH repertoires, where the IGH cDNA libraries were prepared using 5′-RACE-based protocol similar to the protocol used in the current study.

We used repertoires containing more than 5000 clonotypes and processed them in the same way as our data. IGHV gene frequencies were calculated as the number of unique clonotypes to which a particular IGHV gene was annotated by MiXCR divided by the total number of clonotypes identified in the sample. To assess IGHV gene segments over- and under-represented in studied subsets, we utilized edgeR package v0.4.4 (*Robinson et al., 2010*) with the 'trended' dispersion model using trimmed mean of M values method for normalization (*Robinson and Oshlack, 2010*). To evaluate pairwise similarity between repertoires based on IGHV gene segment frequency distributions, we utilized Jensen-Shannon divergence, calculated using the following formula:

$$JS(P,Q) = \frac{1}{2} \sum_i p_i log_2 p_i + \frac{1}{2} \sum_i q_i log_2 q_i - \sum_i \left( \frac{p_i + q_i}{2} log_2 \left( p_i + q_i \right) \right)$$

where $P$ and $Q$ represent distributions of IGHV gene segment in two repertoires, and $p_i$ and $q_i$ represent frequencies of individual member $i$ (IGHV gene segment). In silico repertoires used for the calculation of background clonal overlap (each repertoire contained 5000 clonotypes) were generated with OLGA software v1.0.2 (*Sethna et al., 2019*) under standard settings utilizing the built-in model. For clonal overlap calculation, we downsized repertoires to a fixed number of clonotypes. For *Figure 1B*, the 14,000 most abundant clonotypes were considered in Bmem, 600 in PBL, and 300 in PL. For *Figure 1C*, we considered 5000 clonotypes for all cell subsets. Clonotypes with identical CDR3 amino acid sequence and the same IGHV gene segment detected in both analyzed samples were considered shared. Clonotypes shared between repertoires of at least two individuals were termed as public.

## Assignment of clonal lineages

Change-O v0.4.4 (*Gupta et al., 2015*) was utilized to assign clonal groups, defined as groups of clonotypes with the same V segment, CDR3 length, and at least 85% similarity in CDR3 nucleotide sequence. Before clonal group assignment, we excluded all clonotypes with counts equal to 1. Clonal groups represent observed subsets of clonal lineages originating from a single BCR ancestor, so for simplicity, we use the term 'clonal lineages'. To study evolutionary dynamics of clonal lineages, we joined all replicas, three time points (T1, T2, and T3), and cell subsets for each patient into a single dataset and excluded clonotypes that were presented by a single UMI. Phylogenetic analysis was performed on four patients for whom we had samples at all time points, and on clonal lineages containing at least 20 unique clonotypes as in *Nourmohammad et al., 2019*.

## Clusterization of clonal lineages in HBmem and LBmem clusters

We performed principal component analysis on six scaled variables of clonal lineage composition: fractions of Bmem, PBL, and PL, and fractions of IgM, IgG, and IgA. The IgE isotype was not detected in clonal lineages involved in phylogenetic analysis, so we did not include it as a variable. HBmem and LBmem clusters were defined using the *k*-means clustering algorithm.

## Metric of persistence of clonal lineages

We estimated the frequency of a clonal lineage in the repertoire at a given time point as the ratio of the number of unique clonotypes in the clonal lineage detected at this time point to the overall number of unique clonotypes detected at this time point. If the clonal lineage was not detected at some time point, we assigned its frequency to pseudocount, as it would be a single clonotype detected from this time point. To estimate persistence of clonal lineage frequency in the repertoire over time, we defined the persistence metric:

$$P = \frac{1}{\frac{1}{2}\left(\frac{f_{max}}{f_i} + \frac{f_{max}}{f_j}\right)},$$

where $f_{max}$ is the maximum frequency of the clonal lineage in the three time points and $f_{i,j}$ are its frequencies in the other two (*Figure 3D*). Persistence is equal to 1 if the frequency remains consistent at all three time points. If a clonal lineage was detected just once in the experiment and frequencies at other two time points were assigned to pseudocounts, the persistence approaches zero.

## Reconstruction of clonal lineage germline sequence

We used MiXCR-derived reference V, D, and J segment sequences to reconstruct IGH germline sequences for each clonal lineage, concatenating only those sequence fragments which were present at CDR3 junctions of original MiXCR-defined clonotypes. Thus, random nucleotide insertions were disregarded, making them appear as gaps in the alignment of lineage clonotypes with the germline sequence. We excluded them from all parts of the phylogenetic analysis where germline sequence was required.

## Reconstruction of clonal lineage phylogeny and MRCA

For phylogenetic analysis of clonal lineages, we aligned clonotypes with reconstructed germline sequences using MUSCLE v3.8.31 with 400 gap open penalty (*Edgar, 2004*). Next, we reconstructed the clonal lineage's phylogeny with RAxML v8.2.11, using the GTRGAMMA evolutionary model and germline sequence as an outgroup, and computed marginal ancestral states (*Stamatakis, 2014*). The ancestral sequence of the node closest to the root of the tree, represented by the germline sequence, is the MRCA of the sampled clonotypes. It can match the germline sequence or differ by some amount due to SHM, reflecting the starting point of subsequent evolution of observed clonotypes. This allowed us to distinguish between SHMs fixed in the clonal lineage on the way from the germline sequence to the MRCA (G-MRCA SHMs) vs. polymorphisms within the observed part of lineage. The G-MRCA p-distance in *Figure 4B* was measured as a fraction of diverged positions between germline and MRCA sequences.

## MK test

The MK test is designed to detect the effects of positive or negative selection on population divergence from another species or its ancestral state (**McDonald and Kreitman, 1991**). It is based on the comparison of ratios of nonsynonymous to synonymous substitutions observed in diverged and polymorphic sites, and estimates the fraction of diverged amino acid substitutions fixed by positive selection:

$$\alpha = 1 - \frac{P_n}{P_s} \cdot \frac{D_s}{D_n},$$

where $P_n$ and $P_s$ respectively represent nonsynonymous and synonymous polymorphisms, and $D_n$ and $D_s$ respectively represent nonsynonymous and synonymous divergences fixed in the population. Under neutral evolution, nonsynonymous and synonymous changes are equally likely to be fixed or appear in the population as polymorphisms, so $\frac{D_n}{D_s} = \frac{P_n}{P_s}$ and $\alpha$=0. Positive selection favors adaptive nonsynonymous changes to be fixed, and increases $\frac{D_n}{D_s}$ relative to $\frac{P_n}{P_s}$ , resulting in $\alpha$>0. Negative selection has the opposite effect and produces $\alpha$<0.

To detect selection in the origin of clonal lineages, we considered G-MRCA SHM as divergent changes, and the remaining SHM in a clonal lineage after the MRCA as polymorphic ones (**Figure 5A**). If we observed different nucleotides in the germline sequence and MRCA at a site that was also polymorphic, we considered it as divergent only if the germline variant was not among the polymorphisms (**Figure 5—source data 1**, examples of codons q and r). Codons with unknown germline state were excluded from the MK test (**Figure 5—source data 1**, example of codon j). To perform the MK test on joined HBmem or LBmem cluster variation, we summed variation of all clonal lineages of the same cluster in each category ($D_n$ , $D_s$ , $P_n, P_s$). Calculations of $\alpha$ of distinct clonal lineages for comparison of its distributions between two clusters were complicated by zero G-MRCA distance in some clonal lineages, mostly belonging to the HBmem cluster. We dealt with this using three approaches, presented in **Figure 5—source data 2**. In the first, we added pseudocounts to $D_n$ and $D_s$ in each clonal lineage, so that for clonal lineages with zero G-MRCA distance, $\frac{D_n}{D_s} = 1$. In the second, we excluded clonal lineages with zero G-MRCA distance from the analysis, still adding pseudocounts to $D_n$ and $D_s$ in each clonal lineage in cases where the G-MRCA distance consists of just one nonsynonymous or synonymous substitution. In the third, we compared only those clonal lineages that had at least one nonsynonymous and at least one synonymous substitution on the G-MRCA branch. We also calculated the MK test on joined variation for all types of exclusion criteria to check its robustness; however, there is no need to exclude clonal lineages in the case of the joined test (**Figure 5—source data 2**). In the first approach clonal lineages with zero G-MRCA distance always produced negative $\alpha$ and biased median $\alpha$ to negative values as well. Medians of $\alpha$ in the second and third approaches were more consistent with results of the test on joined variation. However, in the third approach, the filter excluded most of the HBmem cluster, and so in the main test we presented results of the second approach (**Figure 5B**). To check the significance of deviation of $\alpha$ from neutral expectations, we used an Fisher's exact test as in the original MK pipeline (**McDonald and Kreitman, 1991**).

## πNπS

To calculate *πNπS* we identified SHMs in each clonal lineage relative to the reconstructed MRCA sequence. In multiallelic sites (with multiple SHMs observed, see codon *i* in **Figure 5—source data 1** as an example) we considered each variant as an independent SHM event. *πN* and *πS* were calculated as the number of nonsynonymous and synonymous SHMs in a clonal lineage, normalized to the number of nonsynonymous and synonymous sites in the MRCA sequence, respectively. The resulting *πNπS* value is the ratio between *πN* and *πS*:

$$\pi N \pi S = \frac{N}{N_s} : \frac{S}{S_s}$$

where $N$ and $S$ are the number of nonsynonymous or synonymous SHMs, respectively, observed in the clonal lineage and $N_S$ and $S_S$ are the number of nonsynonymous or synonymous sites, respectively, in the MRCA sequence of the clonal lineage, calculated as in **Nei and Gojobori, 1986**.

## Site frequency spectrum

SFS reflects the distribution of SHM frequencies in the clonal lineage. We calculated the frequency of each SHM as a number of unique clonotypes carrying the SHM relative to the overall number of unique clonotypes in the lineage. To visualize SFS, we binned SHM frequencies into 20 equal intervals from 0 to 1 with a step size of 0.05, and counted SHM density in each bin as the number of SHMs in a given frequency bin normalized to the overall number of SHMs detected in the lineage. To obtain the cluster average SFS, we took the mean of clonal lineages of the same cluster in each frequency bin.

## Normalized *πNπS* in bins of SHM frequencies

To compare ratios of nonsynonymous and synonymous SHMs of different frequencies between two clusters, we calculated normalized *πNπS* in bins of SHM frequency. For this purpose we used a smaller number of frequency bins (0; 0.2; 0.4; 0.6; 0.8; 1) to reduce the probability of bins without observed SHMs. To deal with the remaining empty bins, we added pseudocounts to nonsynonymous and synonymous SHMs in each frequency bin. Thus, normalized *πNπS* in the *i*th SHM frequency bin was calculated as follows:

$$\text{normalized } \pi N \pi S = \frac{(N_i+1)/(\sum_{i=1}^{5} N_i+5)/N_S}{N_s}$$

where $N_i$ and $S_i$ are the number of nonsynonymous and synonymous SHMs, respectively, in the *i*th frequency bin, $\sum_{i=1}^{5} N_i$ and $\sum_{i=1}^{5} S_i$ are respectively the overall number of nonsynonymous and synonymous SHMs observed in the clonal lineage (the sum of SHMs in all frequency bins), and $N_S$ and $S_S$ are the number of nonsynonymous and synonymous sites respectively in the MRCA sequence of the clonal lineage, calculated as in *Nei and Gojobori, 1986*. To compare distributions of normalized *πNπS* between two clusters of clonal lineages in the five frequency bins, we used the Mann-Whitney test with Bonferroni-Holm multiple testing correction.

## Acknowledgements

We are grateful to our donors. We are grateful to Alexey Neverov for helpful discussion of inference of selection.

# Additional information

### Funding

| Funder | Grant reference number | Author |
|---|---|---|
| Ministry of Science and Higher Education of the Russian Federation | 075-15-2020-807 | Dmitriy M Chudakov |
| Russian Foundation for Basic Research | 20-34-90153 | Evgeniia I Alekseeva |

The funders had no role in study design, data collection and interpretation, or the decision to submit the work for publication.

### Author contributions

Artem Mikelov, Conceptualization, Data curation, Software, Formal analysis, Validation, Investigation, Visualization, Methodology, Writing – original draft, Writing – review and editing, Participated in study design, Collected cell samples and prepared cDNA libraries, Performed repertoire data processing and analysis; Evgeniia I Alekseeva, Conceptualization, Data curation, Software, Formal analysis, Validation, Investigation, Visualization, Methodology, Writing – original draft, Writing – review and editing, Designed and performed the evolutionary analysis of clonal lineages; Ekaterina A Komech, Writing – review and editing, Contributed to FACS cell sorting experimental design, Assisted with FACS cell sorting experiments; Dmitry B Staroverov, Contributed to FACS cell sorting experimental design, Assisted with FACS cell sorting experiments; Maria A Turchaninova, Writing – review and

editing, Provided advisory support and contributed to the optimisation of IGH library preparation; Mikhail Shugay, Writing – review and editing, Provided advisory support in bioinformatic analysis and result interpretation; Dmitriy M Chudakov, Resources, Funding acquisition, Methodology, Writing – review and editing, Provided advisory support in experimental design and results interpretation; Georgii A Bazykin, Conceptualization, Methodology, Writing – original draft, Writing – review and editing, Designed evolutionary analysis of clonal lineages, Provided advisory support in results interpretation; Ivan V Zvyagin, Conceptualization, Resources, Data curation, Formal analysis, Supervision, Funding acquisition, Validation, Investigation, Visualization, Methodology, Writing – original draft, Project administration, Writing – review and editing, Designed the study, Provided advisory support and participated in cell sample collection

## Author ORCIDs
Artem Mikelov http://orcid.org/0000-0002-1629-2373
Mikhail Shugay http://orcid.org/0000-0001-7826-7942
Dmitriy M Chudakov http://orcid.org/0000-0003-0430-790X
Georgii A Bazykin http://orcid.org/0000-0003-2334-2751
Ivan V Zvyagin http://orcid.org/0000-0002-1769-9116

## Ethics
Human subjects: Informed consent was obtained from each donor. The study was approved by the Local Ethical Committee of Pirogov Russian National Research Medical University, Moscow, Russia (abstract #190 18 Nov 2019).

## Decision letter and Author response
Decision letter https://doi.org/10.7554/eLife.79254.sa1
Author response https://doi.org/10.7554/eLife.79254.sa2

---

# Additional files

## Supplementary files
• MDAR checklist

## Data availability
Sequencing data have been deposited in the ArrayExpress database (www.ebi.ac.uk/arrayexpress, acc. num. E-MTAB-11193). The code for repertoire analysis is available at https://github.com/amikelov/igh_subsets, (copy archived at swh:1:rev:a5cd9753070e319c329ceb4aec8172020ea69138); the code for clonal lineage analysis is available at https://github.com/EvgeniiaAlekseeva/Clonal_group_analysis, (copy archived at swh:1:rev:e14ff814643201ea8278fb51b0f118c869e2dfb9).

The following dataset was generated:

| Author(s) | Year | Dataset title | Dataset URL | Database and Identifier |
|---|---|---|---|---|
| Mikelov AI, Alekseeva EI, Komech EA, Staroverov DB, Turchaninova MA, Shugay M, Chudakov DM, Bazykin GA, Zvyagin IV | 2022 | Longitudinal full-length IGH repertoire profiling and clonal lineage dynamics in memory B cells, plasmablasts and plasma cells of human peripheral blood | https://www.ebi.ac.uk/arrayexpress/experiments/E-MTAB-11193 | ArrayExpress, E-MTAB-11193 |

The following previously published dataset was used:

| Author(s) | Year | Dataset title | Dataset URL | Database and Identifier |
|---|---|---|---|---|
| Gidoni M, Snir O, Peres A, Polak P, Lindeman I, Mikocziova I, Sarna VK, Lundin KEA, Clouser C, Vigneault F, Collins AM, Sollid LM, Yaari G | 2019 | Naive B-cell receptor heavy chain repertoire of celiac patients and healthy controls | https://www.ebi.ac.uk/ena/browser/view/PRJEB26509 | European Nucleotide Archive, PRJEB26509 |

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

## Appendix 1

Four donors (D01, IM, AT, MRK) in our cohort had allergic rhinitis (AR) to pollen. In this note we investigated the influence of donor allergy status on IGH repertoire characteristics described in our study. Peripheral blood samples were collected at three time points: two before pollination season (T1 – March 2017, T3 – March 2018), and one at the peak of birch pollination season in May (T2 – May 2017). Both total and specific IgE serum levels in most AR donors were elevated, while in healthy donors IgE levels were below clinical thresholds (measured with IMMULITE 2000, Siemens). The number of shared clonotypes between donors was not affected by allergy status (*Appendix 1— figure 1*). By focusing on the IgE clonotypes we have not detected shared clonotypes or clonal groups in ASCs subsets of AR donors. We also detected no differences in IGHV gene segment usage between donors depending on allergy status (*Appendix 1—figures 2–4*).

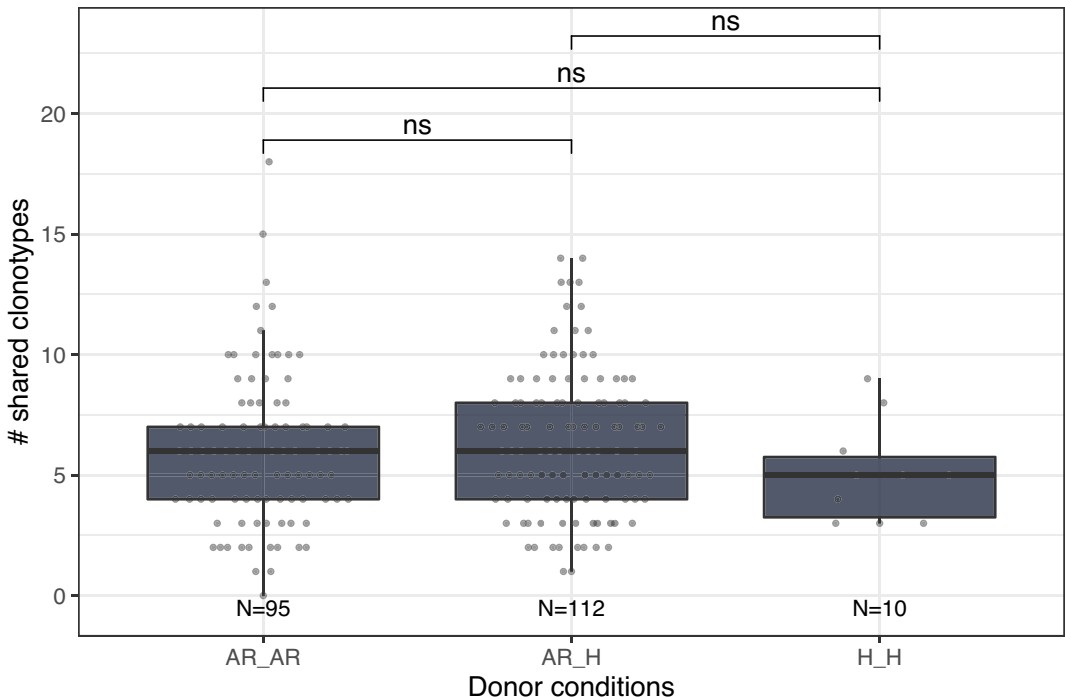

**Appendix 1—figure 1.** Number of shared clonotypes between repertoires of donors without (**H**) or with allergic rhinitis (AR). Each dot represents the number of shared clonotypes between a pair of donors. Comparison between groups was performed using the Mann-Whitney U test. ns corresponds to p ≥ 0.05.

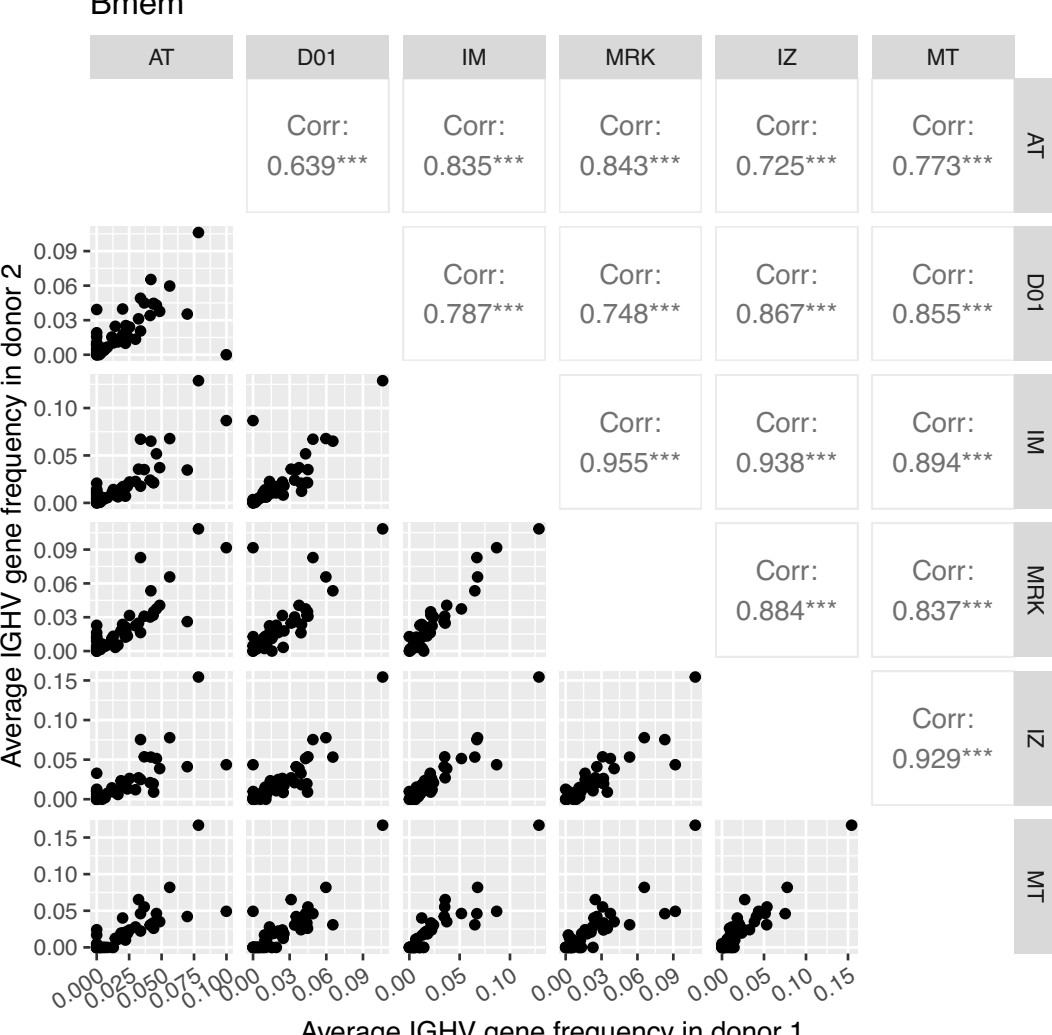

**Appendix 1—figure 2.** Correlation plot of average IGHV gene segment frequencies of memory B cell subset. Down-left part: the dot plots of the average IGHV gene segment frequencies for each pair of donors (axes represent average frequencies, each dot represents a particular IGHV segment for corresponding donors). Top right: Pearson correlation of average frequencies of IGHV gene segment in memory B cell subset between donors, notation of the level of significance is as follows: *=p ≤ 0.05, **=p ≤ 0.01, ***=p ≤ 10⁻³.

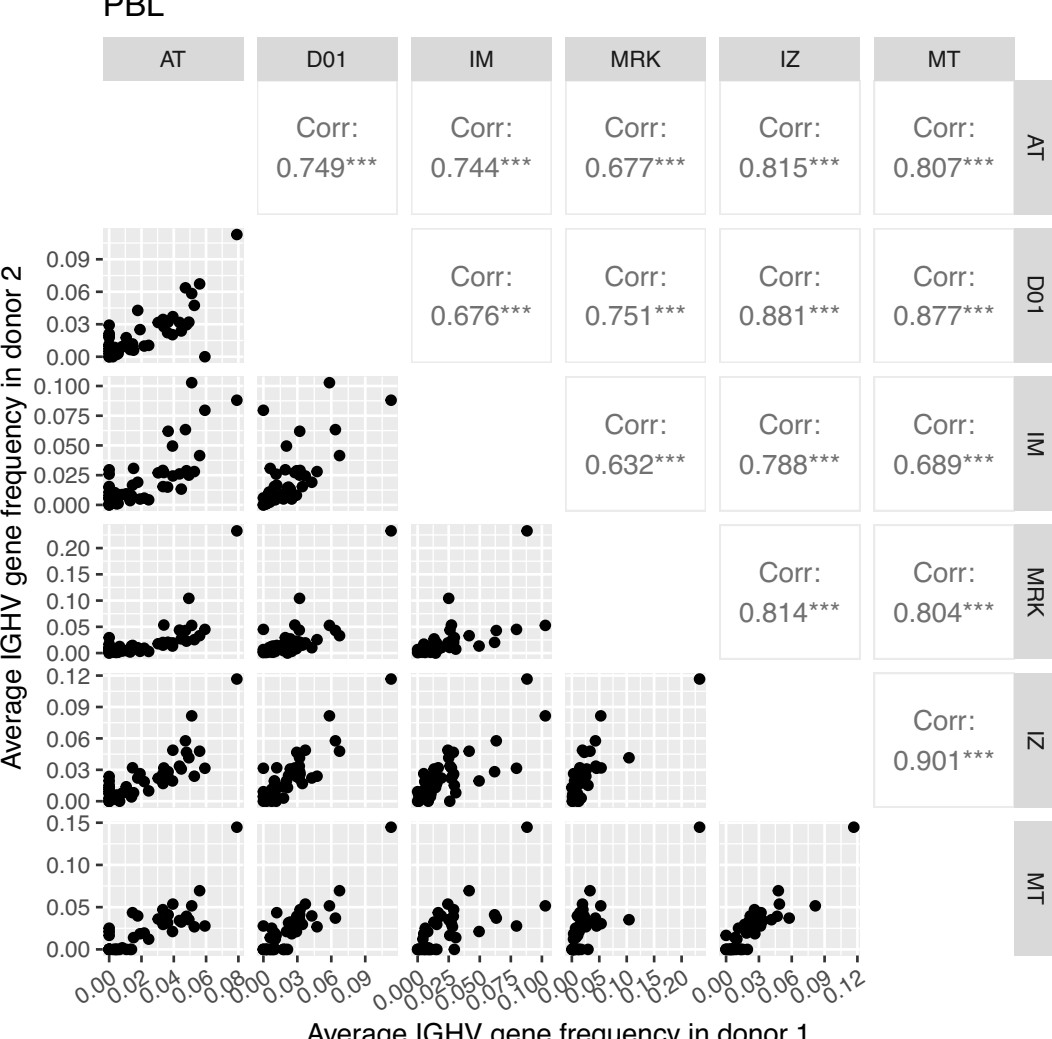

**Appendix 1—figure 3.** Correlation plot of average IGHV gene segment frequencies of the plasmablast subset. Down-left part: the dot plots of the average IGHV gene segment frequencies for each pair of donors (axes represent average frequencies, each dot represents a particular IGHV segment for corresponding donors). Top right: Pearson correlation of average frequencies of IGHV gene segment in plasmablast subset between donors, notation of the level of significance is as follows: *=$p \leq 0.05$, **=$p \leq 0.01$, ***=$p \leq 10^{-3}$.

We observed no effect of pollination season on the dynamics of most abundant clonal lineages as well. Clonal lineages of HBmem cluster had stable repertoire frequency in all three time points in all donors, suggesting no active involvement in ongoing immune response. In contrast LBmem lineages showed features of active immune response: they were composed mostly of ASC clonotypes and exhibited fluctuations in frequency, reflecting changes in the number of detected clonotypes during the observed time period. However, the observed dynamics of LBmem lineages was not clearly correlated with the pollen season in allergic donors (*Figure 3—figure supplement 2*).

Together this data show that the potential reactivation of allergen-specific B cells, which could be expected during pollination season in AR donors, cannot be easily detected on the level of bulk repertoire sequencing. Thus it should not affect the repertoire features of Bmem and ASCs subsets and clonal lineage dynamics. Analysis of allergen-responsive clones and clonal groups requires more precise focusing of sampled B cell subpopulations and sampling time points.

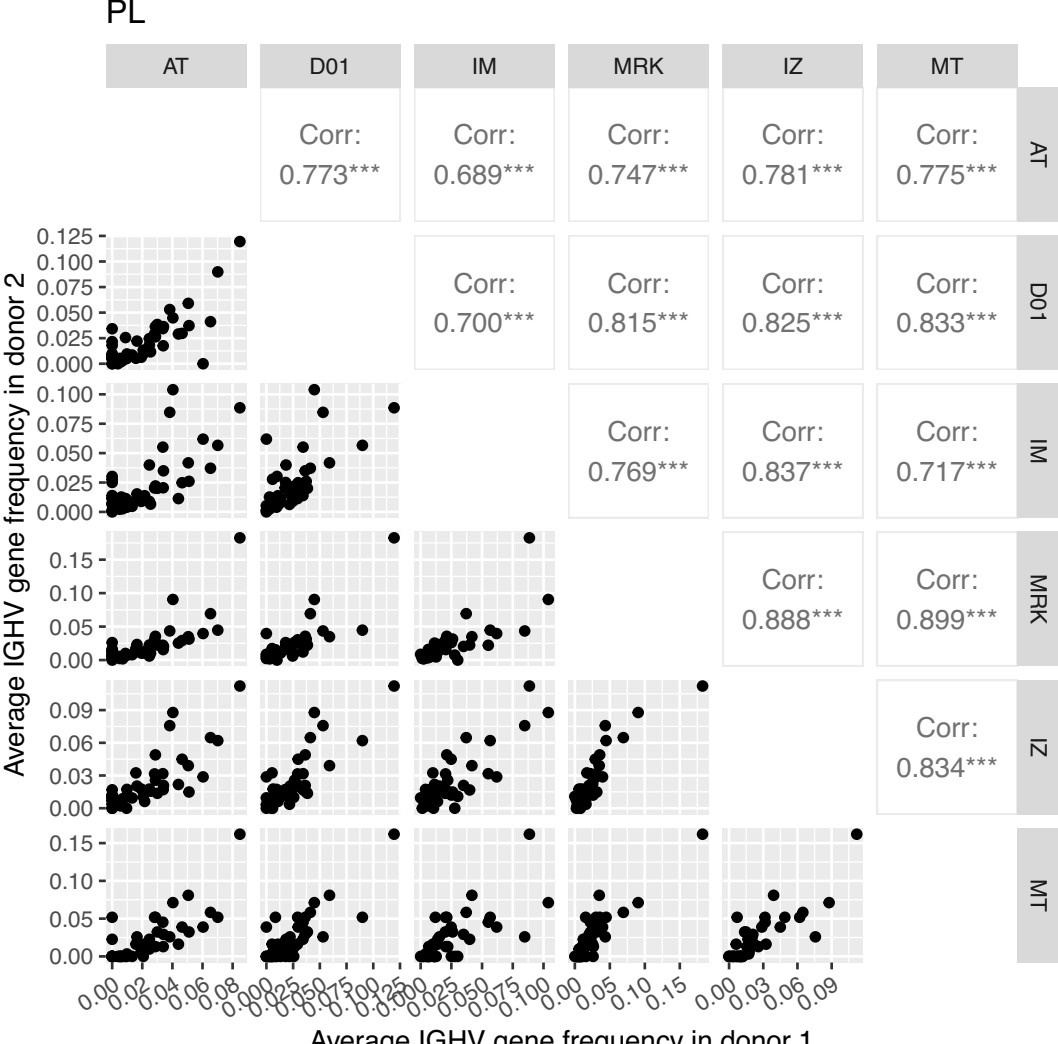

**Appendix 1—figure 4.** Correlation plot of average IGHV gene segment frequencies of plasma cell subset. Down-left part: the dot plots of the average IGHV gene segment frequencies for each pair of donors (axes represent average frequencies, each dot represents a particular IGHV segment for corresponding donors). Top right: Pearson correlation of average frequencies of IGHV gene segment in plasma cell subset between donors, notation of the level of significance is as follows: *=p ≤ 0.05, **=p ≤ 0.01, ***=p ≤ 10⁻³.

