## [Editor Report]

By performing homeostatic longitudinal IgH repertoire analysis of human memory B cells and plasma cells, authors draw two major unique conclusions; first, a high degree of clonal persistence in individual memory B cell subsets with inter individual convergence in memory B cells and plasma cells; second, reactivation of persisting memory B cells with new rounds of affinity maturation during proliferation and differentiation into plasma cells. These conclusions provide significant insight into how human memory B and plasma cells are generated in a homeostatic condition.

---

## [Decision Letter]

**Decision letter after peer review:**

Thank you for submitting your article "Memory persistence and differentiation into antibody-secreting cells accompanied by positive selection in longitudinal BCR repertoires" for consideration by *eLife*. Your article has been reviewed by 2 peer reviewers, and the evaluation has been overseen by a Reviewing Editor and Betty Diamond as the Senior Editor. The following individual involved in review of your submission has agreed to reveal their identity: Wanli Liu (Reviewer #2).

Essential revisions:

1) Throughout the manuscript, the conclusion of each section will help to catch the findings of this study. Please consider including short summaries at the end of each section.

(1.1) p3, line 100; p6, line 179; p17, line 459; p19, line 581; Strictly speaking, "Full-length IgH clonal repertoires" is misleading readers. Indeed, 5'-RACE template switch method adopted by the authors enables to sequence full-part of IGHV segment, however, not for full-length IgH which includes IGHC gene. For accurate descriptions, authors should rephrase these sentences.

(1.2) p4, line 145-147 "The average number of SHMs for IgE clonotypes did not differ significantly between cell subsets but was significantly higher compared to the level of SHM detected for IgM and IgD clonotypes in Bmem."; To compare SHM between IgE and other isotype, it is necessary to compare within the same individual, not between different individuals.

(1.3) p11 Figure 3A; HBmem and LBmem are separated by PC1 score. What is the biological meaning of this PC1 score? That the meaning to divide clonal lineages into two clusters is unclear confuses the interpretation of data to digest.

(1.4) p13 Figure 4F; From the branching data, I was not sure whether lower memory lineage and upper ASC lineage are same origin because branching point could be one or two mutations and if so, how do authors confirm these two lineages are from same origin?

(1.5) p14, line 395-; Probability of nonsynonymous or synonymous mutation will be different based on a kind of original amino acid. Did the authors take into consideration this point to evaluate selection pressure?

2) To calculate background IGHV gene fragments and the number of shared clonotypes, the authors used the data from several database resources. However, the algorithm for normalizing data from all these multiple sources is not elaborated on, and it worried this reviewer that these normalizing criteria shall be uniform for all these databases, otherwise may influence the weight and contribution, and eventually the conclusion.

3) The authors analyzed the distribution frequency of IGHV gene fragments based on Jensen-Shannon divergence. To further validate the conclusion, the authors are encouraged to check if other the index of distributional similarity, for example, Wasserstein distance, will affect the current results.

4) The cohort of 6 healthy donors was not large, and importantly it does not seem to this reviewer that the authors provided details of age, sex, chronic disease, etc. All these messages cannot be missed and instead shall discuss how these factors can influence the conclusion of this manuscript.

*Reviewer #1 (Recommendations for the authors):*

Throughout the manuscript, the conclusion of each section will help to catch the findings of this study. Please consider including short summaries at the end of each section.

p3, line 100; p6, line 179; p17, line 459; p19, line 581; Strictly speaking, "Full-length IgH clonal repertoires" is misleading readers. Indeed, 5'-RACE template switch method adopted by the authors enables to sequence full-part of IGHV segment, however, not for full-length IgH which includes IGHC gene. For accurate descriptions, authors should rephrase these sentences.

p4, line 145-147 "The average number of SHMs for IgE clonotypes did not differ significantly between cell subsets but was significantly higher compared to the level of SHM detected for IgM and IgD clonotypes in Bmem."; To compare SHM between IgE and other isotype, it is necessary to compare within the same individual, not between different individuals.

p11 Figure 3A; HBmem and LBmem are separated by PC1 score. What is the biological meaning of this PC1 score? That the meaning to divide clonal lineages into two clusters is unclear confuses the interpretation of data to digest.

p13 Figure 4F; From the branching data, I was not sure whether lower memory lineage and upper ASC lineage are same origin because branching point could be one or two mutations and if so, how do authors confirm these two lineages are from same origin?

p14, line 395-; Probability of nonsynonymous or synonymous mutation will be different based on a kind of original amino acid. Did the authors take into consideration this point to evaluate selection pressure?

Adding thought of classification of B1-type BCR and B2-type BCR to whole analysis will be interesting and give another layer of understanding of naturally occurring human BCR repertoire.

*Reviewer #2 (Recommendations for the authors):*

There are some points as detailed below that shall be considered before publication.

1) To calculate background IGHV gene fragments and the number of shared clonotypes, the authors used the data from several database resources. However, the algorithm for normalizing data from all these multiple sources is not elaborated on, and it worried this reviewer that these normalizing criteria shall be uniform for all these databases, otherwise may influence the weight and contribution, and eventually the conclusion.

2) The authors analyzed the distribution frequency of IGHV gene fragments based on Jensen-Shannon divergence. To further validate the conclusion, the authors are encouraged to check if other the index of distributional similarity, for example, Wasserstein distance, will affect the current results.

3) The cohort of 6 healthy donors was not large, and importantly it does not seem to this reviewer that the authors provided details of age, sex, chronic disease, etc. All these messages cannot be missed and instead shall discuss how these factors can influence the conclusion of this manuscript.

---

## [Author Response]

Essential revisions:1) Throughout the manuscript, the conclusion of each section will help to catch the findings of this study. Please consider including short summaries at the end of each section.

To address this comment we have added the following conclusions for each section:

lines 177-179: “These observations highlight the differences in general characteristics of IGH repertoire between the Bmem and ASC subsets, and demonstrate similarity of IGHV gene usage that differs from that in naive B cells”.

lines 245-247: “Thus the results demonstrate the level of stability of memory B-cell receptor repertoires and extent of clonal sharing in repertoires of unrelated donors, which might be attributed to exposure to common antigens.”

lines 311-313: “Thus we observed two types of clonal lineages, representing different stages of an immune response: persisting memory with unswitched IgM isotype (HBmem) and responding lineages rapidly increasing in frequency and producing IgG or IgA antibodies (LBmem).”

line 370-375: “To summarize, we observed that LBmem lineages had low level of clonotype divergence and large distance of lineage’s ancestor from the germline sequence, assuming their recent origin from a mature clonotype. The temporal dynamics of LBmem, detection of Bmem clonotypes at the time-point prior to the LBmem lineage expansion, and the relationship between HBmem and LBmem on a clonal lineage level suggest that LBmem expansions may result from reactivation of pre-existing memory.”

line 431-433: “This excess of advantageous SHMs in ancestors of LBmem lineages together with previous observations that LBmem lineages can originate from reactivated memory suggests that reactivation was coupled with new rounds of affinity maturation.”

The concluding summary for the last section was also included in the first version (lines 463-469 in the revised version).

(1.1) p3, line 100; p6, line 179; p17, line 459; p19, line 581; Strictly speaking, "Full-length IgH clonal repertoires" is misleading readers. Indeed, 5'-RACE template switch method adopted by the authors enables to sequence full-part of IGHV segment, however, not for full-length IgH which includes IGHC gene. For accurate descriptions, authors should rephrase these sentences.

We thank the reviewer for this comment, we have corrected the following phrases (the changes we’ve made are highlighted):

line 86 “…that provides full-length IgH variable region sequences with…”lines 98-99 “… we obtained IGH clonal repertoires using a 5’-RACE-based protocol”lines 186-187 “…obtained IGH clonal repertoires by sequencing respective cDNA libraries covering full-length IGH variable domain.”line 484 “Using advanced library preparation technology, we performed a longitudinal study of BCR repertoires…”line 630 “IGH cDNA libraries and sequencing”

(1.2) p4, line 145-147 "The average number of SHMs for IgE clonotypes did not differ significantly between cell subsets but was significantly higher compared to the level of SHM detected for IgM and IgD clonotypes in Bmem."; To compare SHM between IgE and other isotype, it is necessary to compare within the same individual, not between different individuals.

The observation regarding IgE clonotypes was the same for all individual repertoires. To clarify this we have added a new supplementary figure (Supplementary Figure S3 - Figure 1—figure supplement 3) illustrating comparison of SHM rate in Bmem subset between isotypes within each individual. IgE clonotypes, when detected, have on average more SHMs compared with IgM or IgD within the same individual repertoire.

(1.3) p11 Figure 3A; HBmem and LBmem are separated by PC1 score. What is the biological meaning of this PC1 score? That the meaning to divide clonal lineages into two clusters is unclear confuses the interpretation of data to digest.

To describe the clonal relationships between cell subsets we investigated cell subset composition of most abundant clonal lineages. We found that the cell subtypes and isotypes were unequally distributed: some clonal lineages were mostly composed of memory B-cell clonotypes of non-switched isotype IgM, while the others were largely composed of ASCs and enriched in IgG and IgA clonotypes (Supplementary Figure S6B - Figure 3—figure supplement 1B). To understand the nature of such bimodal distribution and compare other features of clonal lineages, differing in cellular and isotype composition, we used PCA and k-mean clustering to split clonal lineages into two clusters based on these features. As expected, fractions of Bmem subset and IgM isotype were main contributors in PC1 score, as shown by arrows on PCA plot (Figure 3A), which represent projections of the corresponding variables onto the two-dimensional PCA plane.

We agree that this logic was presented not clearly enough, so we added a plot with distributions of cell subset and isotype fractions in clonal lineages (Supplementary Figure S6) and modified the narrative in the corresponding paragraph of the text (lines 277-290):

“First we asked how B cell subsets and isotypes were represented in these most abundant clonal lineages. The clonal lineages were mostly composed of memory B-cell clonotypes of non-switched isotype IgM or were largely composed of ASCs, and enriched in IgG and IgA clonotypes (Supplementary Figure S6B - Figure 3—figure supplement 1B). To investigate the nature of such bimodal distribution and perform comparative analysis of these two types of clonal lineages we divided them into two large clusters using k-means clustering algorithm, based on the proportion of represented cell subsets and BCR isotypes(Figure 3A, B, Supplementary Figure S7A - Figure 3—figure supplement 2A). The more abundant HBmem cluster included 138 clonal lineages, and was mostly composed of memory B-cell clonotypes of non-switched isotype IgM. Conversely, the smaller LBmem cluster (52 clonal lineages) was more diverse and largely composed of ASCs, and enriched in IgG and IgA clonotypes. The average size of clonal lineages (*i.e.*, the number of unique clonotypes per lineage) did not differ between the HBmem and LBmem clusters (Supplementary Figure S7B - Figure 3—figure supplement 2B), and both clusters were present in repertoires of all donors (Supplementary Figure S7C - Figure 3—figure supplement 2C).”

Also we mentioned the type of clustering in the caption of the Figure 3A. The former Supplementary Figure S6 was split in two (now Supplementary Figures S6 and S7 - Figure 3—figure supplement 1 and 2) and now includes panel S6B (Figure 3—figure supplement 1B).

(1.4) p13 Figure 4F; From the branching data, I was not sure whether lower memory lineage and upper ASC lineage are same origin because branching point could be one or two mutations and if so, how do authors confirm these two lineages are from same origin?

To address this question we added Supplementary Figure S8 (Figure 4—figure supplement 1) with alignments of CDR regions of clonotypes belonging to the clonal lineage from Figure 4F. Clonal lineages were defined as the group of clonotypes with the same V segment, CDR3 length, and having at least 85% similarity of CDR3 nucleotide sequence. The clonotypes from the lineage shown in Figure 4F passed all of these criteria. The phylogeny for each lineage in the manuscript was reconstructed using the nucleotide sequences of clonotypes in the group.

As it can be seen from the CDR3 nucleotide sequence alignment, of the positions that could be attributed to the non-templated region in D-J junction there is only one position splitting the tree to the two parts belonging to HBmem and LBmem sublineages. At the same time the variations in the rest 12 positions are either distributed between subtrees or conservative. The distribution of nucleotide variants in the other non-templated region (V-D junction) shows the same. On the amino acid level 5 out of 8 positions of CDR3 region encoded by non-templated nucleotides are either conservative or have similar physicochemical properties. Most changes that led to the divergence of ASC sublineage from the remaining tree occurred in CDR2 region, however CDR2s of ASC sublineage clonotypes carry germline amino acid variants in 4 of 7 positions. It can be suggested that 7V and 6Y in CDR2 and CDR3, respectively, were important for differentiation into ASCs.

The position of ASC sublineage on lineage phylogeny also supports that LBmem-like clade has the same origin as the remaining part of clonotypes. Indeed, the LBmem-like clade arises from the middle of the tree (the 4th node from the root), indicating that it has a lot of events in common with the other clades; at the same time the maximum number of nodes from the root to the most distant leaf is 8. It reflects that changes, differing the clade from the remaining clonotypes accumulated gradually and required several branching events. In contrast, when two subgroups are artificially joined together and the sequences within the subgroup are more related to each other than the sequences between the subgroups, one would expect divergence of these subgroups much closer to the root of the phylogeny with rapid appearance of changes in clade sequences relative to the root. We added these arguments in the main text as well:

“Position of ASC sublineage on a distant node from the root of the tree indicates gradual accumulation of SHMs, distinguishing the ASC sublineage from the remaining clonotypes. This fact together with the similarity of CDR3 regions of lineage clonotypes (Supplementary Figure S8 - Figure 4—figure supplement 1) give a reason to conclude that the ASC sublineage has the same origin as the remaining part of the tree with features of HBmem cluster.” (lines 365-375).

(1.5) p14, line 395-; Probability of nonsynonymous or synonymous mutation will be different based on a kind of original amino acid. Did the authors take into consideration this point to evaluate selection pressure?

Yes, we have taken this point into account.To calculate the πNπS we normalized the number of nonsynonymous / synonymous SHMs by the number of nonsynonymous / synonymous sites in the original sequence, which reflects the probability of mutations of a certain type to occur (as specified in Methods section). Number of nonsynonymous / synonymous sites (NS and SS) was calculated according to the classical approach of Nei and Gojobori (Gojobori 1986). In MK test these normalizing numbers are the same for polymorphisms (Pn and Ps) and divergences (Dn and Ds), and become reduced:

α=1 −Pn/SnPs/Ss⋅Ds/SsDn/Sn = 1−PnPs⋅DsDn.

2) To calculate background IGHV gene fragments and the number of shared clonotypes, the authors used the data from several database resources. However, the algorithm for normalizing data from all these multiple sources is not elaborated on, and it worried this reviewer that these normalizing criteria shall be uniform for all these databases, otherwise may influence the weight and contribution, and eventually the conclusion.

For the IGHV gene fragment usage comparison we utilized the trimmed mean of M-values normalization method (TMM) described in Robinson et al. 2010 (DOI: 10.1186/gb-2010-11-3-r25). TMM normalization primarily accounts for library size variation between samples, with regards to the fact that some extremely differentially expressed genes could bias the commonly used normalization procedures (e.g. using total number of reads as normalization factor). We have included the following passages “… data normalization was performed using trimmed mean of M values method (Robinson, 2010a)” and “…using trimmed mean of M values method for normalization (Robinson et al., 2010a).” at lines 196-197 and line 668, correspondingly, to describe the normalization method used. Besides, we highlighted in the Methods section at the lines 659-661 that the data from Gidoni et al. that we used, was obtained using similar cDNA libraries preparation technique (to clarify this we have added the passage *“* where the IGH cDNA libraries were prepared using 5’RACE-based protocol similar to the protocol used in the current study*”*), so we do not expect substantial technical biases in V usage distributions.

Regarding the numbers of shared clonotypes, for each comparison we used the equal number of the clonotypes (5000) – 5000 most abundant clonotypes from Bmem or naive repertoires, 5000 randomly selected from Bmem or 5000 *in silico* generated using OLGA software. We have modified the sentence at line 253-258 to describe it more clearly: “The average number of shared clonotypes between repertoires from pairs of unrelated donors for the most abundant Bmem clonotypes, randomly-selected Bmem clonotypes, most abundant clonotypes from naive repertoires of unrelated donors (from Gidoni et al. 2019), or from synthetic repertoires generated with OLGA software; each repertoire in comparison was represented by a fixed number of clonotypes (5000), either most abundant, randomly selected or generated where indicated.”

3) The authors analyzed the distribution frequency of IGHV gene fragments based on Jensen-Shannon divergence. To further validate the conclusion, the authors are encouraged to check if other the index of distributional similarity, for example, Wasserstein distance, will affect the current results.

To address this point we have tested different metrics of distributional similarity (Jensen-Shannon divergence, 2-Wasserstein distance, correlation of IGHV gene frequencies) for analysis of IGHV gene usage similarity, and obtained the same results. Below we provide distributions of 2-Wasserstein distance calculated in the same way as we did with the Janson-Shannon divergence (Figure 2A).

Distance between repertoires obtained at different time-points from the same or different donors as calculated by 2-Wasserstein distance for IGHV gene frequency distribution. N indicates the number of pairs of repertoires in the group. Comparisons in all panels were performed with two-sided Mann-Whitney U test. * = p ≤ 0.05, ** = p ≤ 0.01, *** = p ≤ 10^-3^, **** = p ≤ 10^-4^.

4) The cohort of 6 healthy donors was not large, and importantly it does not seem to this reviewer that the authors provided details of age, sex, chronic disease, etc. All these messages cannot be missed and instead shall discuss how these factors can influence the conclusion of this manuscript.

We agree with the reviewer on the importance of all the details describing donors in our study. Those details and health status of our donors were provided in Supplementary Table S1. To better describe our cohort we moved the table from supplementary materials to the main text (now named Table 1) and also provided several additions to the Methods section and main text (lines 612-613, 615-616; 94, 119-122).

Four of six donors had food allergy and/or allergic rhinitis, which is the only known chronic condition in this cohort. To better understand whether this condition may affect our observations, we performed an additional comparative analysis of repertoire structure in donors with and without allergic status. The results of the analysis are combined and provided in the Supplementary Note. Besides that we provided the additional panel in Supplementary Figure S7 (see Supplementary Figure S7D - Figure 3—figure supplement 2D), showing that expansion of LBmem clonal lineages is not synchronized in donors with allergic conditions and does not correspond in dynamics to the pollen season. The following phrase has been added at lines 301-302: “The time point of LBmem frequency burst varied between donors (Supplementary Figure S7D - Figure 3—figure supplement 2D)." We also summarized the results of the additional analysis in the Discussion section as well as discussed the limitations of the cohort size (lines 582-600).

Reviewer #1 (Recommendations for the authors):Throughout the manuscript, the conclusion of each section will help to catch the findings of this study. Please consider including short summaries at the end of each section.

To address this comment we have added short summaries to each section (see the Essential Revisions section).

p3, line 100; p6, line 179; p17, line 459; p19, line 581; Strictly speaking, "Full-length IgH clonal repertoires" is misleading readers. Indeed, 5'-RACE template switch method adopted by the authors enables to sequence full-part of IGHV segment, however, not for full-length IgH which includes IGHC gene. For accurate descriptions, authors should rephrase these sentences.

The comment is addressed in the Essential Revisions section.

p4, line 145-147 "The average number of SHMs for IgE clonotypes did not differ significantly between cell subsets but was significantly higher compared to the level of SHM detected for IgM and IgD clonotypes in Bmem."; To compare SHM between IgE and other isotype, it is necessary to compare within the same individual, not between different individuals.

The comment is addressed in the Essential Revisions section.

p11 Figure 3A; HBmem and LBmem are separated by PC1 score. What is the biological meaning of this PC1 score? That the meaning to divide clonal lineages into two clusters is unclear confuses the interpretation of data to digest.

The comment is addressed in the Essential Revisions section.

p13 Figure 4F; From the branching data, I was not sure whether lower memory lineage and upper ASC lineage are same origin because branching point could be one or two mutations and if so, how do authors confirm these two lineages are from same origin?

The comment is addressed in the Essential Revisions section.

p14, line 395-; Probability of nonsynonymous or synonymous mutation will be different based on a kind of original amino acid. Did the authors take into consideration this point to evaluate selection pressure?

The comment is addressed in the Essential Revisions section.

Adding thought of classification of B1-type BCR and B2-type BCR to whole analysis will be interesting and give another layer of understanding of naturally occurring human BCR repertoire.

We thank the reviewer for the idea, it would be of great interest to add analysis of B1-/B2-type BCRs to the picture. However, the current ambiguity in B1/B2 characteristics does not allow to associate a clonotype with B1/B2 subset just on the basis of BCR sequence. We believe that such attempt should be accompanied with sorting of particular populations as done by Rodriguez-Zhurbenko et al. (DOI: 10.3389/fimmu.2019.00483), where authors sequenced repertoires of sorted human B1-cells to investigate age-associated alterations.

Reviewer #2 (Recommendations for the authors):There are some points as detailed below that shall be considered before publication.1) To calculate background IGHV gene fragments and the number of shared clonotypes, the authors used the data from several database resources. However, the algorithm for normalizing data from all these multiple sources is not elaborated on, and it worried this reviewer that these normalizing criteria shall be uniform for all these databases, otherwise may influence the weight and contribution, and eventually the conclusion.

The comment is addressed in the Essential Revisions section.

2) The authors analyzed the distribution frequency of IGHV gene fragments based on Jensen-Shannon divergence. To further validate the conclusion, the authors are encouraged to check if other the index of distributional similarity, for example, Wasserstein distance, will affect the current results.

The comment is addressed in the Essential Revisions section.

3) The cohort of 6 healthy donors was not large, and importantly it does not seem to this reviewer that the authors provided details of age, sex, chronic disease, etc. All these messages cannot be missed and instead shall discuss how these factors can influence the conclusion of this manuscript.

The comment is addressed in the Essential Revisions section.